# Unraveling the plasticity of translation initiation in prokaryotes: Beyond the invariant Shine-Dalgarno sequence

Karel Estrada[1,2], Alejandro Garciarrubio[3], Enrique Merino[4]*

1 Centro de Investigación en Dinámica Celular, Instituto de Investigación en Ciencias Básicas y Aplicadas, Universidad Autónoma del Estado de Morelos (UAEM), Cuernavaca, Morelos, México, 2 Massive Sequencing and Bioinformatics Unit, Instituto de Biotecnología, Universidad Nacional Autónoma de México, Cuernavaca, Morelos, México, 3 Department of Cell Engineering and Biocatalysis, Instituto de Biotecnología, Universidad Nacional Autónoma de México, Cuernavaca, Morelos, México, 4 Department of Molecular Microbiology, Instituto de Biotecnología, Universidad Nacional Autónoma de México, Cuernavaca, Morelos, México

☉ These authors contributed equally to this work.
* enrique.merino@ibt.unam.mx

**Data Availability Statement:** All relevant data are within the paper and its Supporting information files.

## Abstract

Translation initiation in prokaryotes is mainly defined, although not exclusively, by the interaction between the anti-Shine-Dalgarno sequence (antiSD), located at the 3'-terminus of the 16S ribosomal RNA, and a complementary sequence, the ribosome binding site, or Shine-Dalgarno (SD), located upstream of the start codon in prokaryotic mRNAs. The antiSD has a conserved $5'-CCUCC-3'$ core, but inter-species variations have been found regarding the participation of flanking bases in binding. These variations have been described for certain bacteria and, to a lesser extent, for some archaea. To further analyze these variations, we conducted binding-energy prediction analyses on over 6,400 genomic sequences from both domains. We identified 15 groups of antiSD variants that could be associated with the organisms' phylogenetic origin. Additionally, our findings revealed that certain organisms exhibit variations in the core itself. Importantly, an unaltered core is not necessarily required for the interaction between the 3'-terminus of the rRNA and the region preceding the AUG of the mRNA. In our study, we classified organisms into four distinct categories: i) those possessing a conserved core and demonstrating binding; ii) those with a conserved core but lacking evidence of binding; iii) those exhibiting binding in the absence of a conserved core; and iv) those lacking both a conserved core and evidence of binding. Our results demonstrate the flexibility of organisms in evolving different sequences involved in translation initiation beyond the traditional Shine-Dalgarno sequence. These findings are discussed in terms of the evolution of translation initiation in prokaryotic organisms.

## Introduction

The start of an mRNA's translation depends on several primary and secondary RNA structure elements. These include the start codon type [1], the absence of RNA secondary structures [2],

**Funding:** The funders had no role in study design, data collection and analysis, decision to publish, or preparation of the manuscript.

**Competing interests:** The authors have declared that no competing interests exist.

the presence of an S1 ribosomal protein [3, 4], the presence of a 5' untranslated leader region in the mRNA [5], and a ribosome binding site or the Shine-Dalgarno sequence (SD) [6]. When present, the SD is generally located around 8 bases upstream of the start codon (AUG). While SD sequences are typically variable, they are partially complementary to the antiSD in the 16S rRNA of the species. On the other hand, the antiSD is very similar between species. It typically lies a few nucleotides before the 3'-end of the rRNA, overlapping a very conserved CCUCC motif (the antiSD "core"). The interaction usually involves 4 to 8 bases, with 3 or more belonging to the core. Thermodynamically, the strength of interaction between the SD and the antiSD is measured as a change in Gibbs free energy (deltaG or dG, for short). It is measured in kcal/mol units, and the more negative the value, the stronger the interaction. It is known that stronger interactions allow higher rates of translation initiation. The SD's interaction with the antiSD is vital in most prokaryotic organisms for establishing the correct translation initiation site [7].

However, Bacteroidetes, a bacterial lineage, generally does not use this process [4, 8, 9]. Instead, they utilize other mRNA elements for initiation, including a specific upstream set of adenines [10]. Even though Bacteroidetes have an conserved antiSD, their ribosomes do not recognize SD sequences [11]. A study by McNutt et al [12] showed that the antiSD's role in *Flavobacterium johnsoniae*, a Bacteroidetes member, is significantly less relevant than in *Escherichia coli*, aligning with the two organisms' SD usage.

In *Flavobacterium johnsoniae*'s ribosomes, the antiSD sequence is concealed by proteins (bS21, bS18, and bS6) due to uniquely conserved amino acids [9]. This suggests a common mechanism across the Bacteroidetes class. Interestingly, if the bS21 protein is eliminated, the translation of reporter genes with an SD increases in *Flavobacterium johnsoniae* [13].

The *rpsU* gene in *F. johnsoniae* and related bacteria, which encodes bS21, contains a strong SD sequence [11]. This indicates that the SD sequence could be used as a regulatory element, enabling these proteins to control their translation. A proposed model suggests that as bS21 levels increase, complete ribosomes will also increase, consequently reducing bS21 production [11], creating an auto-regulatory mechanism. This correlates with research showing that bS21 protein synthesis depends on a bS21-lacking ribosome subpopulation [13].

Like bacteria, many archaeal genes use the SD-antiSD mechanism. However, many archaeal mRNAs are "leaderless," having a 5' untranslated region (UTR) shorter than 8 nucleotides [14]. These mRNAs do not have an SD sequence, so the ribosome directly recognizes the initiation codon (AUG) [15]. Previously, it was thought that archaea relied on RNA/RNA interaction for translation, but recent studies suggest that archaea may use two different initiation mechanisms: one bacterial-like based on SD motifs and another for translating leaderless mRNAs [16].

In Archaea, SD motifs are essential for translation initiation, with their removal potentially stopping protein production. Still, even with the 5' UTR removed, the mRNA could be translated, indicating a different mechanism [17]. It is suggested that this mechanism might involve ribosomal subunits pre-loaded with initiator tRNA, similar to eukaryotic initiation. Archaea's small ribosomal subunits could interact strongly with leadered mRNAs if SD motifs are present but not with leaderless ones, requiring met-tRNAi presence for initiation site recognition.

In the leaderless translation process, the initiation factor 2 (IF2) plays a crucial role. In bacteria, IF2 promotes the entry of the initiator tRNA (f-met-tRNA) into the ribosomal P site. For leaderless translation, 30S ribosomal subunits pre-loaded with initiator tRNA are required [18]. Leaderless mRNAs recognize a 5' terminal AUG codon by interacting with the ribosome-bound f-met-tRNA anticodon without any prior mRNA/ribosome interaction. IF2 is central to this process, similar to eukaryotic initiation. The abundance of IF2, especially the IF2/IF3 ratio, favors "leaderless" translation over leadered, SD-based translation [19]. Thus, the IF2 factor is vital to the initiation process in the leaderless translation mechanism.

Considering the different pathways for the translation initiation process in prokaryotic organisms and the current greater representativeness of sequenced organisms, our study aimed to analyze the bases in the 16S rRNA and mRNAs involved to define the translation start site. Although the comparison of SD and antiSD sequences among a set of model organisms has been documented [8, 20, 21], in most of these studies, the SD sequences of organisms have been defined based on over representation of some nucleotide sequences found a short distance upstream of the mRNA AUG initiation codon and, a posteriori, the verification of a complementary sequence in the 16S rRNA 3'-terminus, without considering the stabilization energy of the base-pairing in such interactions. Considering the above, the objective of the present study was to carry out a statistical analysis of the thermodynamic interactions between the 3'-terminus of 16S rRNA and the 5′ UTR of mRNAs that would better define the most likely antiSD and SD sequences in the set of genomic sequences currently available to observe the characteristics of these sequences specific for each prokaryotic phylogenetic class.

We comprehensively analyzed SD:antiSD interactions in a set of over 6,400 non-redundant genome sequences from the KEGG Database [22] to identify potential variations from those defined for model organisms that could be related to their phylogenetic origins. Unlike previous studies that primarily focused on bacteria [1, 23–30], we extended our investigation to include both bacteria and archaea. Our results indicate that the most frequent bases for the interaction vary between different lineages, both in composition (given interspecies variation in the rRNA 3;-end), in width, and in their location relative to the CCUCC motif. (In the few organisms where the CCUCC is not perfectly conserved, a probable position was inferred by alignment to the very strong consensus of the region in most species). By summarizing at the taxonomic level of class, we discovered 15 different types of antiSD from Bacteria and Archaea.

Besides discovering class specific antiSD, our results show that there are 4 types of organisms. The first type represents the majority of organisms. These organisms possess the CCUCC motif in their rRNA and demonstrate frequent SD:antiSD interactions, which involve a significant proportion of the genes. The second type also has the CCUCC motif in their rRNA; however, SD:antiSD interactions in these organisms are rare. Interestingly, the third group of organisms lacks the conserved CCUCC motif, yet they still show a high degree of SD-antiSD interaction. Lastly, there is a small set of organisms that neither have the CCUCC motif nor display a significant amount of SD:antiSD interaction.

The results obtained in our study broaden our vision of translation initiation in prokaryotic organisms and allow us to recognize the plasticity of RNA elements to interact and participate in this essential molecular process.

## Materials and methods

### Organisms included in this study

We selected 6,457 organisms from the 7,272 prokaryotic genomes present in the KEGG database [22] as of May 2022 for our study. These organisms, outlined in S1 Table, were chosen based on their genomic annotations, which exhibited the highest number of CDS, and had at least one 16S rRNA with a full 3' terminal region. These organisms represent 87 phylogenetic classes, both from Bacteria and Archaea.

### Retrieval of 16S rRNAs and 5' untranslated mRNA sequences

The anti-Shine-Dalgarno sequence lies in the unstructured rRNA region found between the furthest downstream hairpin of the RNA and the 3'-end point of the RNA. The exact 3'-end nucleotide of the 16S rRNA genes was obtained using Ribbon [31], a tool designed to curate

16S genes. Ribbon uses the MEME/MAST software to anchor the sequence's end. This is a brief description of the actual steps: a) We generated a representative sequence motif for the 3'-termini of prokaryotic 16S rRNAs, which includes the antiSD and conserved flanking sequences. This was achieved by randomly selecting 280 16S rRNA sequences from representative bacterial organisms and 33 16S rRNA sequences from representative archaeal organisms used in our study. b) These sequences of approximately 1,500 base pairs in length were analyzed using the MEME program [32] to identify conserved sequence motifs and their corresponding MEME matrices using the parameters "-motifs" set to 50, "-evt" set to 1e-50 and "-w" set to 60 (S1 Fig). c) The MEME matrix corresponding to all non-variable regions of 16S rRNA, which commonly includes the CCUCC of the antiSD in the last motif, was used with the MAST program [33, 34] to differentiate which 16S rRNA genes of the studied organisms contained the antiSD region and which did not. The parameters used in our MAST program were "-ev" set to 1e-100, "-mt" set to 1e-08, and "-best". Ribbon uses position-dependent probabilities and is able to find the 3'end of the 16S gene regardless of the presence of the conserved CCUCC motif.

To determine the last hairpin secondary structure and the downstream unstructured region, we performed a secondary structure analysis of the last 40 nucleotides of the 16S rRNAs using RNAfold from the Vienna RNA Package [35] with default parameters. With only 13 exceptions, the unstructured 3'-terminus region of the 16S rRNAs was 15 bases long for Bacteria and 13 bases long for Archaea. The sequence of the unstructured region for each organism is included in S2 Table.

The 5' untranslated region (see Fig 1) of the mRNAs were obtained by considering only the protein-coding genes as annotated in the KEGG database. We used the annotated start codons as references without any further verification. Alternative start codons were not tried. While annotation errors on the rRNA ends could have an important impact on our results, annotation errors in the UTRs, if few, would have minimal impact since they are averaged over hundreds or thousands of genes.

## Verification of the CCUCC core in the 16S rRNA unstructured regions

Based on the identification of the unstructured region, the presence/absence of the antiSD core sequence CCUCC was inspected by carefully examining the multiple sequence alignment

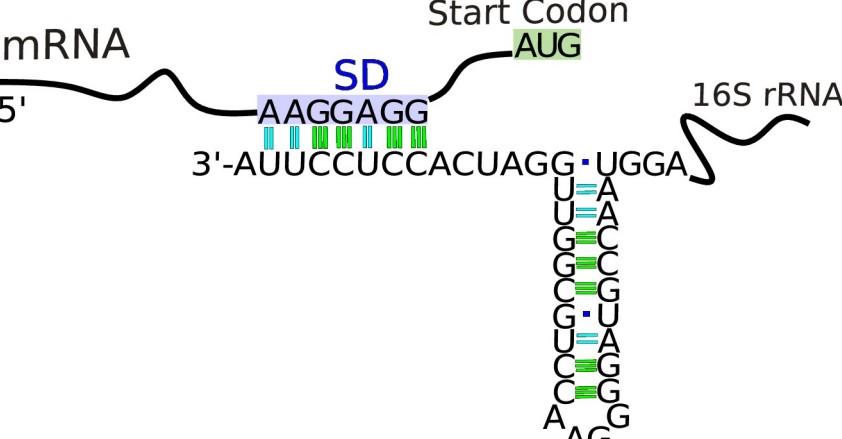

**Fig 1. The 3'-terminus of the 16S rRNAs capable of interacting with the 5' UTR region of mRNAs is unstructured.** The secondary structure of the last 40 nucleotides of the E. coli 16S rRNA is presented. The antiSD core, the SD sequence, and the start codon are colored red, purple, and green, respectively. Note that the last 15 nucleotides of the 16S rRNA are unstructured and correspond to the most likely region to interact with the SD sequence in the mRNAs.

of the last 25 bases of these sequences generated by the Mafft program [33] using its default parameters. In addition, for organisms with more than one 16S rRNA gene, we verified the presence/absence of the CCUCC in all their paralogous copies. When a correctly aligned CCUCC was not found, we verified if the variation is typical of the lineage by comparing with the closest organisms. We aligned the unstructured region of all rRNAs lacking a conserved CCUCC with the unstructured region of *E. coli* using the Needle program from EMBOSS Suite (http://emboss.open-bio.org/) and inspected each alignment manually. As a final verification, we retrieved the genomic sequence of these organisms from NCBI (https://www.ncbi.nlm.nih.gov/genome/) to see if the genome in that database had a CCUCC, in contrast with the genome from KEGG. Only after conducting all these analyses, if an organism still lacked a CCUCC motif, could we categorize it as a variant without a conserved CCUCC.

### Identification of SD and antiSD sequences by hybridization of the 16S unstructured regions with the mRNA UTRs

To analyze the SD:antiSD interactions, we considered the 18 nucleotides long region (here called UTR) located from -20 to -2 bases upstream of the start codon of the mRNAs. The free energy of hybridization calculation was performed using the RNAhybrid software [36] using the default parameters. RNAhybrid ignores intramolecular structures, and all multi-branch loops search for thermodynamically favorable interaction regions and offer a detailed output of the interacting nucleotides (Fig 2). We ran RNAhybrid over all 10 nts long subsequences of the UTR (a sliding window), and the most favorable interaction (lowest deltaG) from the 9 windows was used as the match. Matches with a dG < -8.4 kcal/mol were considered to be Shine-Dalgarno (SD) sequences. Following Starmer, J., et al (2006) [37], this dG threshold was determined as the average dG of the 4 tetramers that can match the conserved CCUCCUU 16S core: AAGG (-7.1 kcal/mol), AGGA (-8.0 kcal/mol), GGAG (-9.3 kcal/mol) and GAGG (- 9.1 kcal/mol), as reported by RNAhybrid.

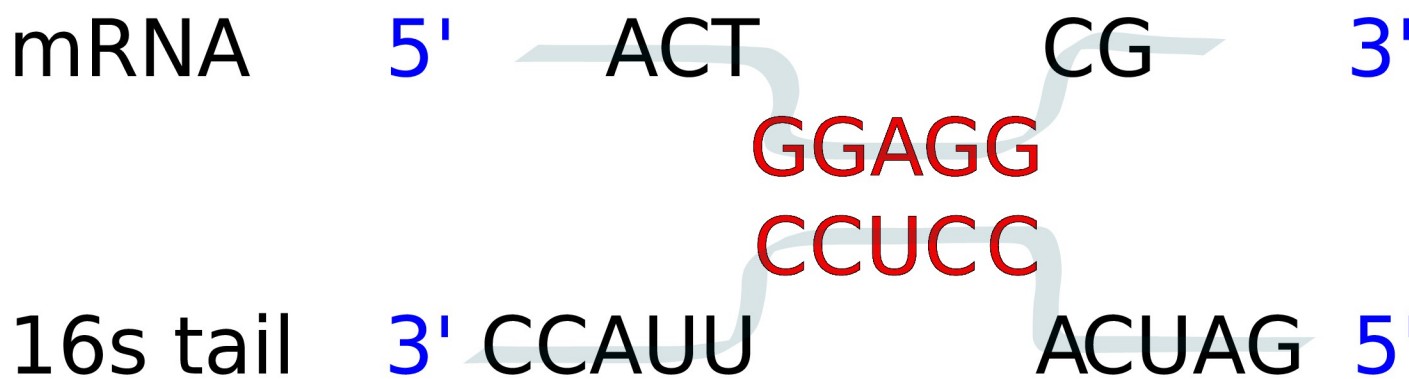

**Fig 2. Example of the output of the RNAhybrid program.** The interacting bases in the SD and antiSD sequences are colored in red. Note that -13.2 kcal/mol corresponds to the most significant value for a perfect hybridization of the canonical core of the SD and antiSD sequences.

## Construction of antiSD profiles

For every SD, the positions and the interacting bases in the SD:antiSD match were registered to evaluate their relative frequencies. The pair G:U was also considered; it is the only non-Watson-Crick pair used by RNAhybrid. These frequencies were estimated considering: a) all the mRNA of an organism; and b) all the mRNAs of all the organisms of a particular phylogenetic class. To determine the sequence profile of an antiSD, we built a histogram of the frequencies. All positions whose frequency was above 0.4 times that of the mode were considered part of that antiSD profile. Profiles were compared by sequence, width, and position, which is relative to the central U in the CCUCC motif. For better visualization and analysis, the frequencies of interacting nucleotides at a given position in the antiSD and SD sequences are shown as histograms. We only created antiSD profiles for those organisms whose frequency of SD was above 5.5% relative to the total number of mRNAs. Organisms with a lower SD percentage were considered "SD-Scarce": poor users of the SD:antiSD mechanism. We evaluated the impact of various percentage thresholds on the classification of organisms utilizing the SD:ASD mechanism. We discovered that, exclusively within the 4–7% range, any chosen threshold resulted in almost identical classifications, indicating a degree of stability. This suggests that any threshold within this range would be less arbitrary. Consequently, we considered the threshold of 5.5% to be a reasonable value for our study.

To determine the antiSD profile of a phylogenetic class, we calculated the base frequencies for all members of the class with their cognate antiSD, as described above, and mapped them to the most common antiSD in the class.

## Phylogeny of selected organisms representing different types of antiSD and SD sequences

To analyze the phylogenetic distribution of the organisms with the different characteristics of their SD and antiSD sequences, we build a phylogenetic tree that includes (1) a random sample of 252 organisms from the list of representative organisms considered in our study and (2) 71 representative organisms that lack a conserved CCUCC motif in their 16S rRNA gene. The complete list of organisms used in the phylogeny is shown in the S3 Table. Subsequently, the SSU-align (v0.1.1) [38] program [38] was used to align the 16S rRNA sequences, using the parameter "--key-out default" that causes pre-calculated masks to be used in the 16S rRNA of bacteria and archaea for discarding the uninformative positions of multiple alignments.

To construct the phylogeny, we first used the Jmodeltest program (v2.1.10) [39] with the parameters "-i -f -g 4 -t ML -AIC -BIC ". This allowed us to select the optimal model. We retained the model recommended by both the Akaike and Bayesian criteria. Thus, we ran the Phyml program (v20120412) [40] with the following parameters "-d nt -n 1 -b 0—run_id GTR + I + G -m 012345 -fm -ve -c 4 -ae—-no_memory_check -o tlr -s BEST". Finally, visualization and tree editing were performed using iTOL [41].

## Availability of Perl scripts

The Perl scripts used to process the RNAhybrid results and generate the histograms of the relative frequencies at which the SD:antiSD sequence bases interact, considering representative organisms by species and classes, can be found at https://github.com/kjestradag/antiSD_paper_scripts.

## Results and discussion

### Analysis of the number of paralogous copies of the 16S rRNA per organism

It has been documented that the number of paralogous copies of rRNA genes in bacterial organisms can vary significantly. Although the reasons for this variation are not entirely clear, it has been proposed that the number of rRNA gene copies is associated with an adaptive response of bacteria to their corresponding ecological niches, nutrient availability, and genome sizes. In our case, from the set of 6,457 organisms, 5,396 of them were found to have more than one copy of the 16S rRNA gene (Fig 3); nevertheless, in more than 98% of cases, we identified that the last twenty bases of the 16S rRNA gene paralogous copies were identical. Moreover, in almost all organisms with parallel copies of the 16S rRNA gene, most copies conserve the CCUCC at their 3'-terminus. Whenever possible, thermodynamic hybridization analyses were performed with such 16S rRNAs.

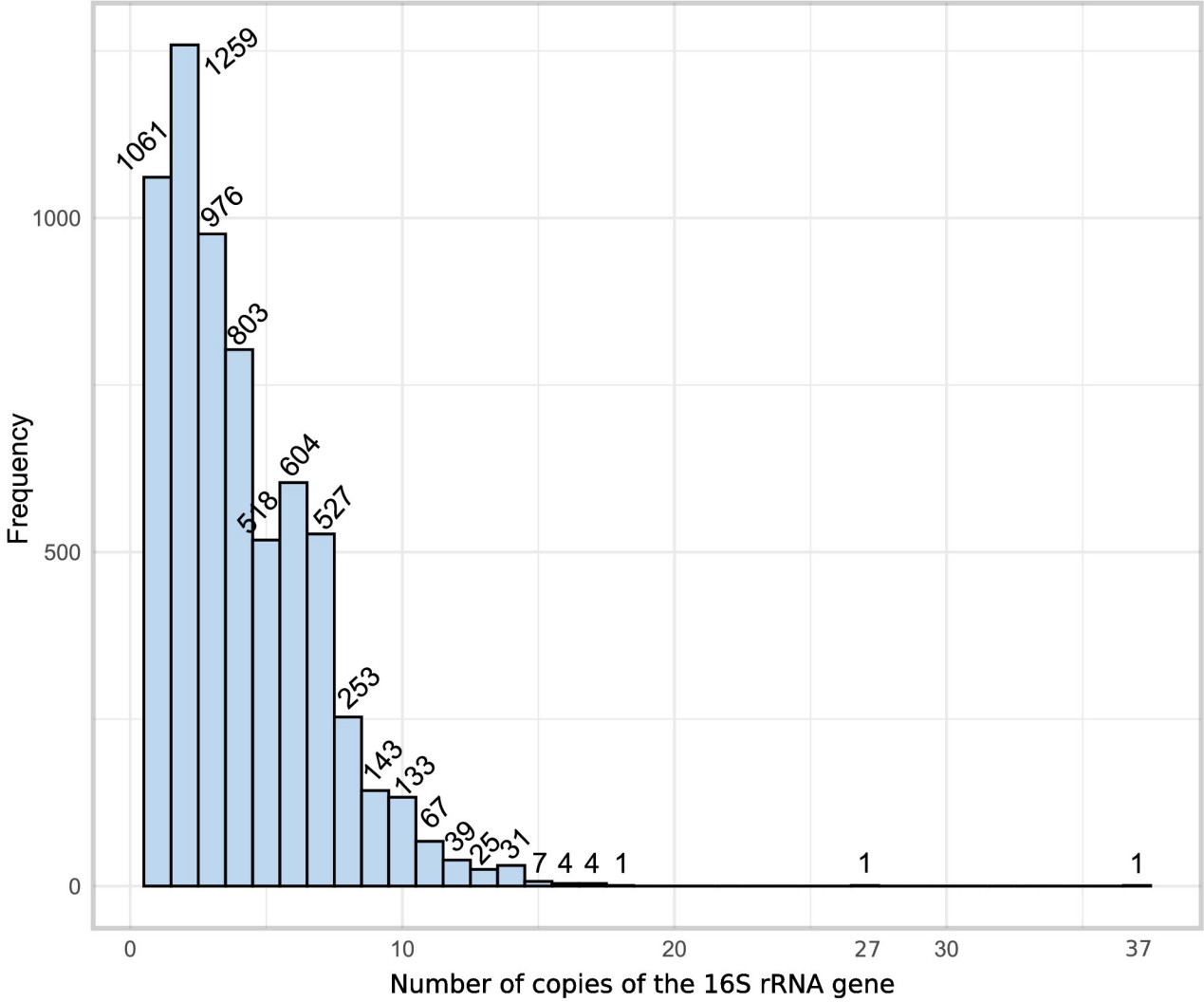

**Fig 3. Distribution of the 16S rRNA copy number in the study organisms.** 16S rRNA genes were identified in our set of 6,457 selected organisms using the MAST program with MEME position-probability matrices built considering a set of representative sequences of 16S rRNA genes from bacteria and archaea, as described in the Materials and Methods section.

## Curation of rRNA 3'-end and identification of the unstructured region

The antiSD is located in the unstructured region (UR) of the rRNA, specifically between the furthest downstream stem-and-loop secondary structure and the termination point of the RNA (Fig 1). The final nucleotide of the rRNAs was determined with Ribbons [31], a tool designed to improve the annotation of rRNA genes. To find the unstructured region, we predicted secondary structures in the last 40 nucleotides of the rRNA using the RNAfold program [35]. With only 13 exceptions in our collection of 6457 organisms, the length of the unstructured region was 15 nts for Bacteria and 13 nts for Archaea. The nucleotide sequences of this unstructured 3'-terminus of our studied organisms are shown in S2 Table.

## Analysis of the SD and antiSD of model organisms

In the first instance, we used our bioinformatic protocol on the model organisms *Escherichia coli* and *Bacillus subtilis* since the SD and antiSD of these organisms have been recently analyzed [21]. An UTR was defined as the region from -20 to -2 upstream from the start codon. Although we did not curate the annotated positions of start codons, we believe a few wrong positions would have minimal impact as they are averaged with many more correct ones. The dG of interaction between an UTR and the unstructured region of the rRNA was predicted with RNAhybrid [42]. If the strongest match in an UTR had an energy of -8.4 kcal/mol or better (more negative), the gene was assumed to have a functional SD. The selection of -8.4 kcal/mol as the threshold is justified in Material and Methods. Our study found that 64.2% of the *E. coli* mRNAs have an SD that hybridizes with the cognate antiSD at -8.4 or less kcal/mol (S2 Table). This frequency is considerably smaller than that of *B. subtilis*, where 87.4% of the genes have such an interaction (S2 Table). Moreover, we observed that the antiSD sequences of these organisms, shown in Fig 4, presented significant differences regarding their length (7 vs. 10),

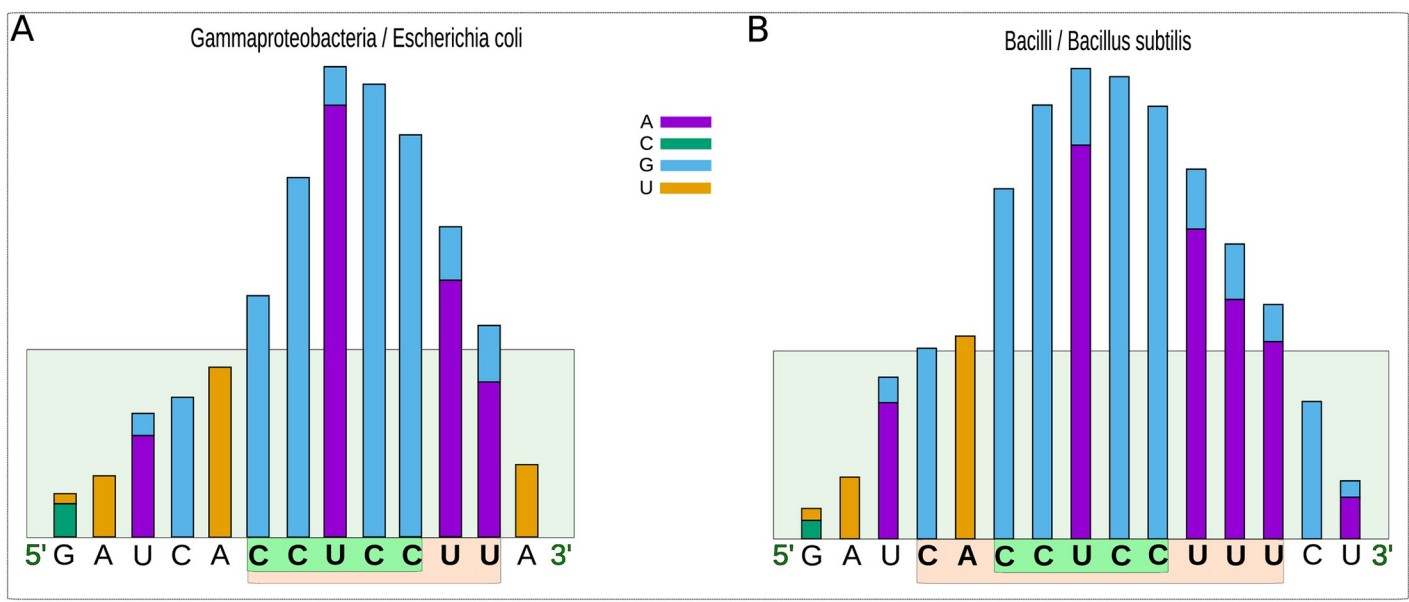

**Fig 4. Histograms of the relative frequencies at which the antiSD/SD sequence bases interact in model organisms.** Based on our thermodynamic analyses, we predicted the bases of the leader sequences of the mRNAs in an organism most likely to interact with the 3'-terminus of their 16S rRNAs. We then represented the frequencies of the interacting bases as histograms. Positions with a relative frequency greater than 40% (green area up) were considered part of the SD sequence, and their counterpart in the 16S rRNA was considered an antiSD sequence. At each position, the frequency of the interacting base in the mRNA is indicated. The histograms illustrate SD:antiSD interactions for the model organisms *E. coli* (A) and *B. subtilis* (B). The *E. coli* antiSD sequence (5'-CCUCCUU-3') and the *B. subtilis* extended antiSD sequence (5'-UCACCUCCU-3') are shown in orange. The canonical CCUCC is shown in green.

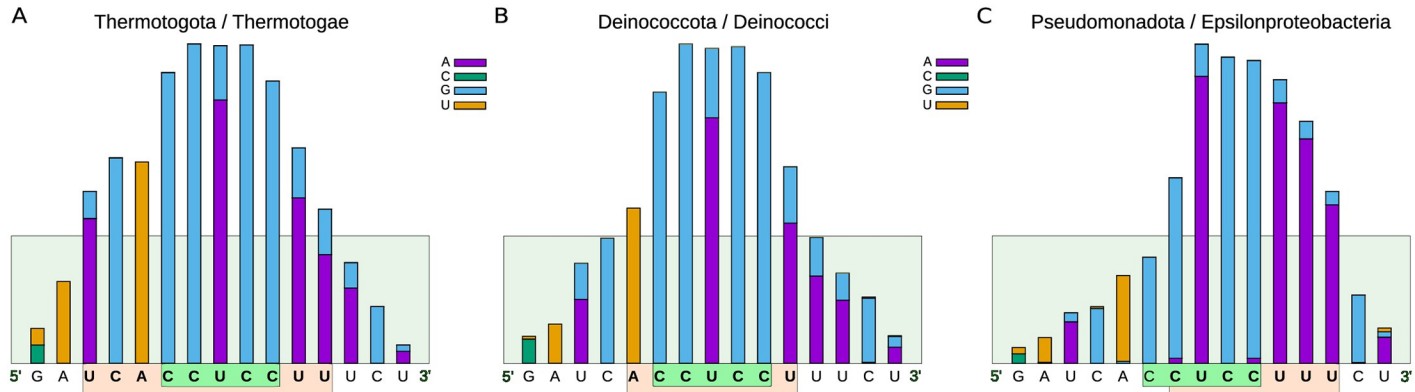

**Fig 5. Histograms of the relative frequencies at which the nucleotides of the antiSD and SD interact in model-representative phylogenetic classes.** (A) Thermatogae, (B) Acidobacteria, and (C) Epsilonproteobacteria, as examples of phylogenetic classes in which histograms of the relative frequencies of SD:antiSD interacting bases are skewed to the left, centered, or skewed to the right. Their antiSD sequences and canonical CCUCC are shown in orange and green, respectively.

sequence (CCUCCUU vs. CACCUCCUUU), and their precise position from the 3' end of the 16S rRNAs. In *E. coli*, the interaction occurs between nucleotides -8 to -2, while in *B. subtilis*, it occurs between nucleotides -12 to -3, from the 3'-end nucleotide. The antiSD sequences of these and the rest of the organisms included in our study are shown in S2 Table.

## Analysis of the types of antiSD by phylogenetic classes

To obtain better-supported and statistically significant results on the definitions of the antiSD and SD consensus sequences, we repeated the analytical procedure described above for the complete set of representative sequence genomes grouped by phylogenetic classes. As a result, out of the 87 classes initially considered in our research, 78 use the 16S rRNA/mRNA interaction to define the translation start site and conserve the CCUCC in at least 5.5% of the genes, which was our cutoff criterion. The histogram analysis reveals the diversity of the antiSD elements in terms of their position relative to the CCUCC, which can be linked to the phylogenetic origin of the microorganisms. As examples of this diversity, Fig 5 shows the frequency histograms of three representative classes, Epsilonproteobacteria, Thermatogae, and Deinococci, with different tendencies to be skewed to the right, left, or center, respectively. Preliminary results indicate that the mean position of the antiSD profile in an organism is poorly correlated with the mean position of SD (S2 Fig).

In addition to the general differences observed in the histograms of being biased towards one of their extremes, left, centered, or right, we carried out a more detailed study in which we considered both the relative position of the antiSD in terms of the core element CCUCC as well as the differences in the neighboring sequences. Based on these variations, we divided the phylogenetic classes analyzed into fifteen groups (Fig 6). Notably, the antiSD and SD of the model organism E. coli, where pioneering work was done to define the interactions between 16S rRNA and mRNAs, correspond to the second largest group in our list of antiSD/SD types (Group 8, Fig 6).

The analysis of the antiSD sequence variations, including their sequence and length, indicates a correlation with the phylogenetic origin of the microorganisms. Fig 7 presents representative histograms from each of our fifteen groups. One of the more evident variations has been previously reported for phylogenetic classes belonging to the Archaea domain (groups 12 to 15); group 12 includes classes from the phyla Crenarchaeota, Thaumarchaeota, and

```
Group   #Classes
  1        3      ........CCUCCUUUC   ⊕
  2        1      .........CUCCUUU.   ⊕
  3       23      .......CCUCCUUU.    ⊕
  4        9      ......ACCUCCUUU.    ⊕
  5        2      ....UCACCUCCUUU.    ⊕
  6        4      ....UCACCUCCUU..    ⊕
  7        2      .....CACCUCCUU..    ⊕
  8       11      .......CCUCCUU..    ⊕
  9        4      ......ACCUCCUU..    ⊕
 10        3      ......ACCUCCU...    ⊕
 11        2      .....CACCUCCU...    ⊕
 12        8      ....UCACCUCCU...    ⊕
 13        1      ...AUCACCUCCU...    ⊕
 14        4      ....UCACCUCC....    ⊕
 15        1      ...AUCACCUC.....    ⊕
                             ⊕Bacteria ⊕Archaea
```

**Fig 6. Classification of antiSD sequences in accordance with their sequence and position in the 16S rRNA 3'-terminus.** Based on the frequency of interactions between the nucleotides of the 16S rRNA 3'-terminus and the bases of the 5' upstream regions of mRNAs, we defined and classified the antiSD sequences in our set of reference organisms according to their position relative to the CCUCC, which is present in almost all 16S rRNA molecules. The letters in the figure represent the consensus of interacting bases in the rRNA of each group, while dots indicate non-interacting bases. The consensus sequences can be found in Fig 7. The CCUCC is enclosed in a red rectangle. Pie charts show the relative frequencies of bacteria and archaea in each group.

Euryarchaeota, while groups 13 to 15 only include classes from the phylum Euryarchaeota, in which histograms are shifted to the left concerning the histogram of group 3 that clusters most of the organisms with "classical" antiSD/SD sequences. Table 1 compiles the antiSD and SD consensus sequences of our studied organisms, the category to which they belong, and the percentage of mRNAs that, according to our theoretical analysis, possess a functional SD sequence. It is important to note that the values of these percentages vary significantly from classes almost lacking SD sequences in their mRNAs to classes with up to 86% of their messengers having SD sequences. Our results agree with previously reported frequencies of SD for phylogenetic clades in which organisms possess bona fide SD sequences in their mRNAs [43] and for some archaeal organisms in which the translation initiation process is independent of an SD sequence [1, 44].

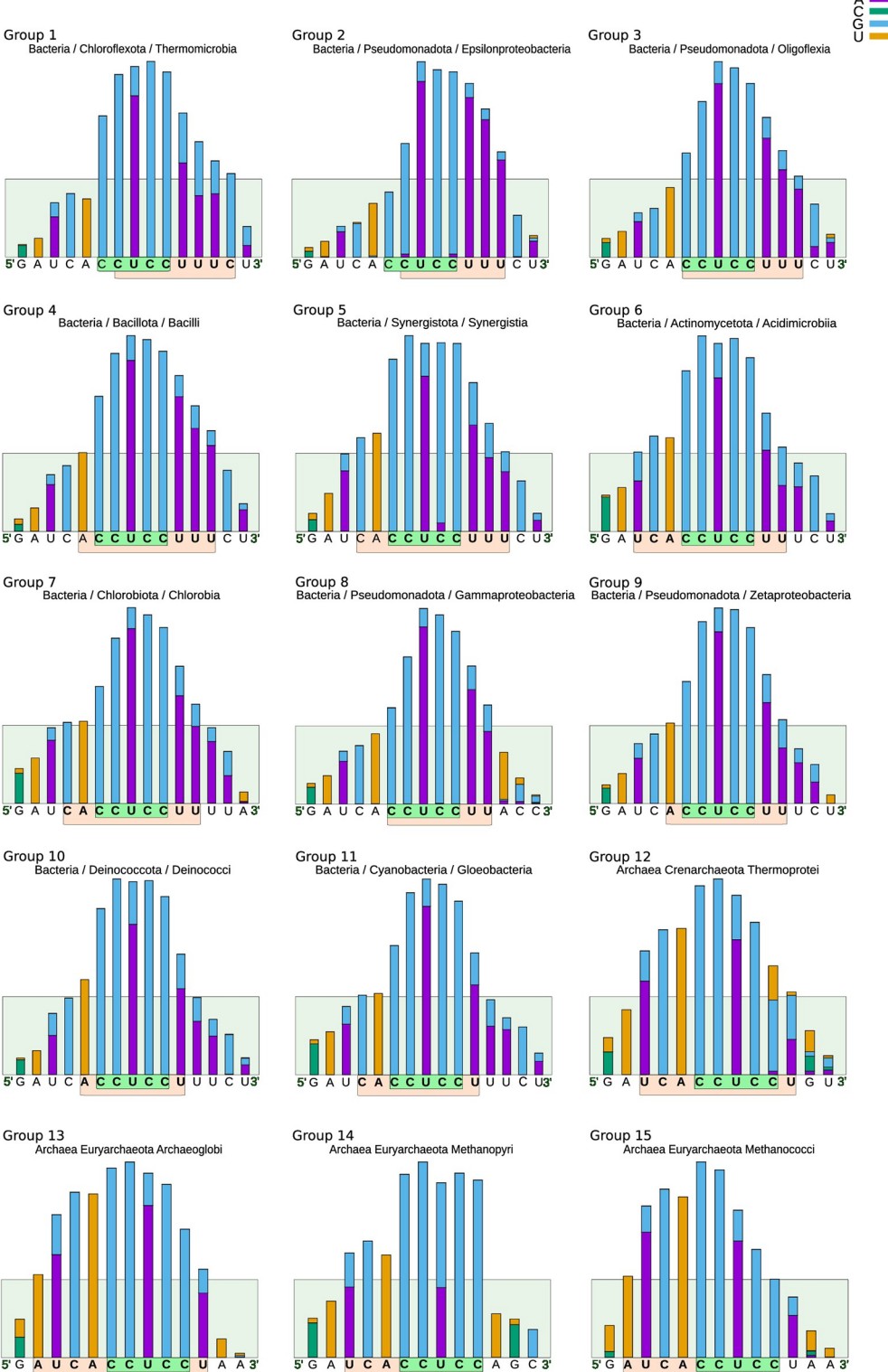

**Fig 7. Histograms of representative classes of the fifteen groups of antiSD identified in our analysis.** Of the 78 phylogenetic classes, one is selected from each of the ten antiSD classes we defined based on the similarity of their sequences, lengths, and positions within their 3'-terminus.

**Table 1. SD and antiSD sequences of organisms grouped by phylogenetic classes.**

| G | D | Phylum | Class | % SD | antiSD Pos. | antiSD Length | antiSD (5'-3') | SD (5'-3') |
|---|---|--------|-------|------|-------------|---------------|----------------|------------|
| 1 | Bacteria | Actinomycetota | Coriobacteriia | 74 | -2+6 | 9 | CCUCCUUUC | GAAAGGAGG |
|   |   | Chloroflexota | Ktedonobacteria | 64 |   |   |   |   |
|   |   |   | Thermomicrobia | 78 |   |   |   |   |
| 2 |   | Pseudomonadota | Epsilonproteobacteria | 51 | -1+5 | 7 | CUCCUUU | AAAGGAG |
| 3 |   | Acidobacteriota | Acidobacteriia | 45 | -2+5 | 8 | CCUCCUUU | AAAGGAGG |
|   |   | Actinomycetota | Rubrobacteria | 58 |   |   |   |   |
|   |   |   | Actinomycetia | 51 |   |   |   |   |
|   |   | Armatimonadota | Chthonomonadetes | 70 |   |   |   |   |
|   |   | Calditrichaeota | Calditrichae | 51 |   |   |   |   |
|   |   | Chloroflexota | Chloroflexia | 61 |   |   |   |   |
|   |   |   | Anaerolineae | 58 |   |   |   |   |
|   |   |   | Caldilineae | 42 |   |   |   |   |
|   |   | Elusimicrobiota | Elusimicrobia | 46 |   |   |   |   |
|   |   |   | Endomicrobia | 43 |   |   |   |   |
|   |   | Fibrobacterota | Fibrobacteria | 37 |   |   |   |   |
|   |   | Fusobacteriota | Fusobacteriia | 79 |   |   |   |   |
|   |   | Ignavibacteriota | Ignavibacteria | 32 |   |   |   |   |
|   |   | Kiritimatiellaeota | Kiritimatiellae | 63 |   |   |   |   |
|   |   | Lentisphaerota | Lentisphaeria | 60 |   |   |   |   |
|   |   | Mycoplasmatota | Mollicutes | 41 |   |   |   |   |
|   |   | Nitrospinota | Nitrospinia | 37 |   |   |   |   |
|   |   | Nitrospirota | Nitrospira | 57 |   |   |   |   |
|   |   | Planctomycetota | Phycisphaerae | 46 |   |   |   |   |
|   |   | Pseudomonadota | Hydrogenophilalia | 58 |   |   |   |   |
|   |   |   | Oligoflexia | 47 |   |   |   |   |
|   |   |   | Alphaproteobacteria | 47 |   |   |   |   |
|   |   | Verrucomicrobiota | Methylacidiphilae | 15 |   |   |   |   |
| 4 |   | Acidobacteriota | Holophagae | 61 | -3+5 | 9 | ACCUCCUUU | AAAGGAGGU |
|   |   | Bacillota | Negativicutes | 81 |   |   |   |   |
|   |   |   | Tissierellia | 78 |   |   |   |   |
|   |   |   | Clostridia | 78 |   |   |   |   |
|   |   |   | Bacilli | 74 |   |   |   |   |
|   |   |   | Erysipelotrichia | 74 |   |   |   |   |
|   |   | Dictyoglomota | Dictyoglomia | 64 |   |   |   |   |
|   |   | Nitrospirota | Thermodesulfovibrionia | 45 |   |   |   |   |
|   |   | Thermodesulfobacteriota | Thermodesulfobacteria | 50 |   |   |   |   |
| 5 |   | Synergistota | Synergistia | 83 | -5+5 | 11 | UCACCUCCUUU | AAAGGAGGUGA |
|   |   | Verrucomicrobiota | Spartobacteria | 10 |   |   |   |   |
| 6 |   | Actinomycetota | Acidimicrobiia | 45 | -5+4 | 10 | UCACCUCCUU | AAGGAGGUGA |
|   |   |   | Thermoleophilia | 45 |   |   |   |   |
|   |   | Coprothermobacterota | Coprothermobacteria | 63 |   |   |   |   |
|   |   | Thermotogota | Thermotogae | 80 |   |   |   |   |
| 7 |   | Acidobacteriota | Blastocatellia | 32 | -4+4 | 9 | CACCUCCUU | AAGGAGGUG |
|   |   | Chlorobiota | Chlorobia | 13 |   |   |   |   |

(*Continued*)

**Table 1.** (Continued)

| G | D | Phylum | Class | % SD | antiSD Pos. | antiSD Length | antiSD (5'-3') | SD (5'-3') |
|---|---|---|---|---|---|---|---|---|
| 8 |  | Actinomycetota | Nitriliruptoria | 51 | -2+4 | 7 | CCUCCUU | AAGGAGG |
|  |  | Bacillota | Limnochordia | 78 |  |  |  |  |
|  |  | Chlamydiota | Chlamydiia | 34 |  |  |  |  |
|  |  | Chloroflexota | Dehalococcoidia | 59 |  |  |  |  |
|  |  | Chrysiogenota | Chrysiogenetes | 74 |  |  |  |  |
|  |  | Planctomycetota | Planctomycetia | 48 |  |  |  |  |
|  |  | Pseudomonadota | Deltaproteobacteria | 59 |  |  |  |  |
|  |  |  | Acidithiobacillia | 57 |  |  |  |  |
|  |  |  | Gammaproteobacteria | 47 |  |  |  |  |
|  |  |  | Betaproteobacteria | 37 |  |  |  |  |
|  |  | Spirochaetota | Spirochaetia | 46 |  |  |  |  |
| 9 |  | Aquificota | Aquificae | 48 | -3+4 | 8 | ACCUCCUU | AAGGAGGU |
|  |  | Caldiserica | Caldisericia | 53 |  |  |  |  |
|  |  | Deferribacterota | Deferribacteres | 61 |  |  |  |  |
|  |  | Pseudomonadota | Zetaproteobacteria | 56 |  |  |  |  |
| 10 |  | Chloroflexota | Tepidiformia | 50 | -3+3 | 7 | ACCUCCU | AGGAGGU |
|  |  |  | Ardenticatenia | 44 |  |  |  |  |
|  |  | Deinococcota | Deinococci | 45 |  |  |  |  |
| 11 |  | Acidobacteriota | Vicinamibacteria | 41 | -4+3 | 8 | CACCUCCU | AGGAGGUG |
|  |  | Cyanobacteria | Gloeobacteria | 37 |  |  |  |  |
| 12 |  | Armatimonadota | Fimbriimonadia | 25 | -5+3 | 9 | UCACCUCCU | AGGAGGUGA |
|  |  | Gemmatimonadota | Gemmatimonadetes | 22 |  |  |  |  |
|  |  | Thaumarchaeota | Cyanophyceae | 36 |  |  |  |  |
|  | Archaea | Euryarchaeota | Methanobacteria | 56 |  |  |  |  |
|  |  |  | Methanomicrobia | 46 |  |  |  |  |
|  |  | Thaumarchaeota | Conexivisphaeria | 36 |  |  |  |  |
|  |  |  | Nitrososphaeria | 11 |  |  |  |  |
|  |  | Thermoproteota | Thermoprotei | 31 |  |  |  |  |
| 13 |  | Euryarchaeota | Archaeoglobi | 34 | -6+3 | 10 | AUCACCUCCU | AGGAGGUGAU |
| 14 |  | Euryarchaeota | Thermococci | 73 | -5+2 | 8 | UCACCUCC | GGAGGUGA |
|  |  |  | Methanopyri | 75 |  |  |  |  |
|  |  |  | Nanohaloarchaea | 22 |  |  |  |  |
|  |  |  | Halobacteria | 23 |  |  |  |  |
| 15 |  | Euryarchaeota | Methanococci | 69 | -6+1 | 8 | AUCACCUC | GAGGUGAU |

The frequencies of the interacting bases between the 16S rRNA 3'-terminus and the immediate upstream region of the AUG of the mRNAs of the organisms included in our study were determined and used to define the antiSD consensus sequence of the organisms grouped by phylogenetic classes. In "antiSD Pos." column, the first and last nucleotides of the antiSD sequence are measured from the central U nucleotide of the CCUCC, which is position 0. In the table header, "G" stands for groups and "D" stands for domain.

It is worth noting that, although group 12 includes phylogenetic classes from bacteria and archaea, they share the same antiSD profile that, compared to other groups, is longer and shifted to the left relative to the CCUCC. The most representative classes of this group are Fimbriimonadia, Gemmatimonadetes and Cyanophyceae from the Bacteria domain, and Methanobacteria, Methanomicrobia, Conexivisphaeria, Thermoprotei, and Nitrososphaeria from the Archaeal domain (Fig 8). These results indicate that instead of there being one consensus

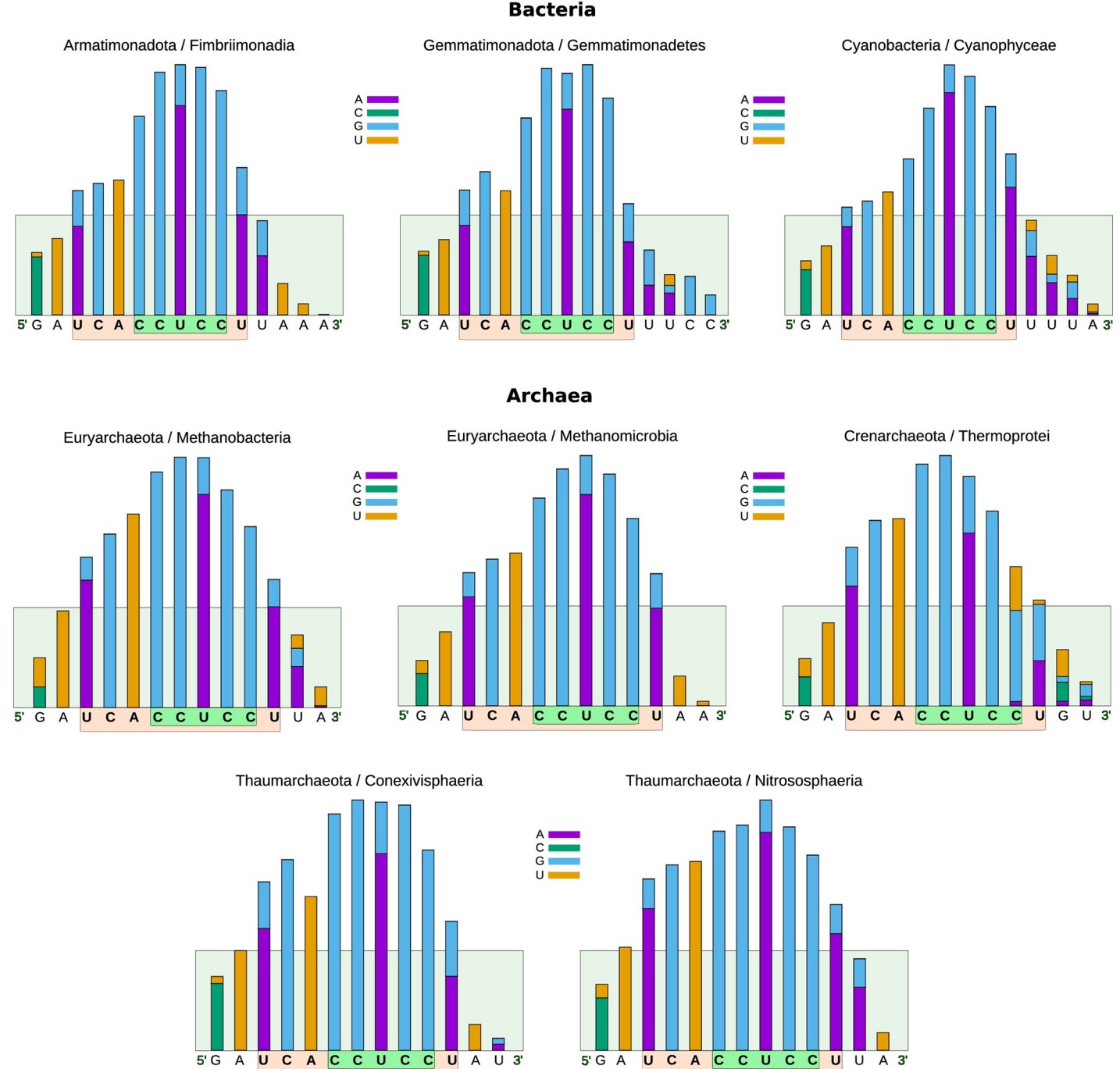

**Fig 8. Breakdown of the phylogenetic classes in group 8 (from Fig 7).** The histograms include three phylogenetic classes of archaea and five of bacteria. Group 8 is the only one that includes members from both domains. This group is characterized by having a relatively long antiSD sequence that starts further left from the CCUCC, compared to most bacterial antiSD sequences.

sequence for bacteria and another for archaea, as previously reported [44], there is a broader set of antiSD/SD sequences that organisms have selected to define the translation initiation sites and whose differences are associated with the phylogenetic classes to which they belong.

### Classification of organisms according to the presence/absence of the antiSD sequence and the ability of their mRNAs to interact with the 16S rRNA

Each of the previously mentioned 15 groups, classified based on their antiSD signature, corresponds to taxonomic classes that fulfill two requirements:

1. They exhibit a recognizable canonical CCUCC.

2. They have a predicted SD in 5.5% or more of their mRNAs.

However, not all taxonomic classes fulfill these conditions [2, 7, 9]. For our subsequent analysis, an organism is considered to have the CCUCC if all five bases are conserved and is considered to be "SD-Abundant" if more than 5.5% of the mRNAs demonstrate SD:antiSD binding. Combining the presence or absence of these two characteristics, we can classify all organisms into four categories, as described next. The phylogenetic distribution of these categories is shown in Fig 9.

**i) Organisms with high-occurrence of SD and conserved CCUCC.**   This category is the largest one. From our set of 6,457 organisms considered in our study, 92.6% fall into this category (Fig 9, non-colored organisms). This category comprises the 78 classes with assigned antiSD profiles (Fig 6). Although the criterion for being "SD-Abundant" is 5.5% of genes with SD, most organisms in this category have a much higher frequency. The Bacteria *E. coli* and *B. subtilis*, which were our reference species, belong here. They have 64% and 84% of genes with SD, respectively. The category includes the majority of Bacterial and Archaeal species. Interestingly, the halophilic Archaea (Halobacteria), which initiate the translation of many of their genes with an SD-independent mechanism, also belong here. The SD:antiSD interaction in this category is considered canonical since it encompasses most organisms with the broadest phylogenetic distribution. Almost 50 years ago, the SD sequence, the complementarity with the 16S rRNA end, and the role in translation initiation were first proposed by J. Shine and L. Dalgarno [6] in a study involving 4 members of this category: *E. coli*, *P. aeruginosa*, *B. stearothermophilus*, and *C. crescentus*.

**ii) "SD-Scarce" organisms with conserved CCUCC.**   Despite possessing the canonical CCUCC in their rRNAs, organisms in this group mostly do not form stable interactions with the 5' region preceding the AUG start codon. These organisms include mainly members of the classes Bacteroidia, Flavobacteriia, Chitinophagia, Cytophagia, and Sphingobacteriia of the phylum Bacteroidota, but also Mollicutes of the phylum Mycoplasmatota. To a lesser extent, we also identify in this second group some members of the Verrucomicrobia phylum, Akkermansia muciniphila and Verrucomicrobia bacterium IMCC26134, one Gammaproteobacteria, and an Archaea organism, Nanoarchaeum equitans Kin4-M from the Nanoarchaeales class. These organisms are shown as yellow-colored nodes in Fig 9. The low percentage of their SD:antiSD interactions is indicated by the small height of the orange bars in the figure. Our study confirms and expands on earlier findings for certain species in these classes, which are known to initiate translation independently of SD:antiSD interactions [8]. Using a foreign SD, McNutt [13] has shown that the antiSD of *F. johnsoniae*, a member of this category, is functional for SD:antiSD pairing as long as the bS21 protein is absent. However, our results would predict that even if the antiSD was not sequestered by bS21, most of the mRNAs of *F. johnsoniae* would interact poorly with the antiSD. Considering that the initiation of translation in these organisms is independent of the direct interaction between the 16S rRNA and the leader region of the mRNA, as well as their evolutionary closeness to many other species lacking antiSD, it is possible to think that these species are in transition toward the final loss of the antiSD nucleus and will eventually become part of our fourth category, shown below. The S2

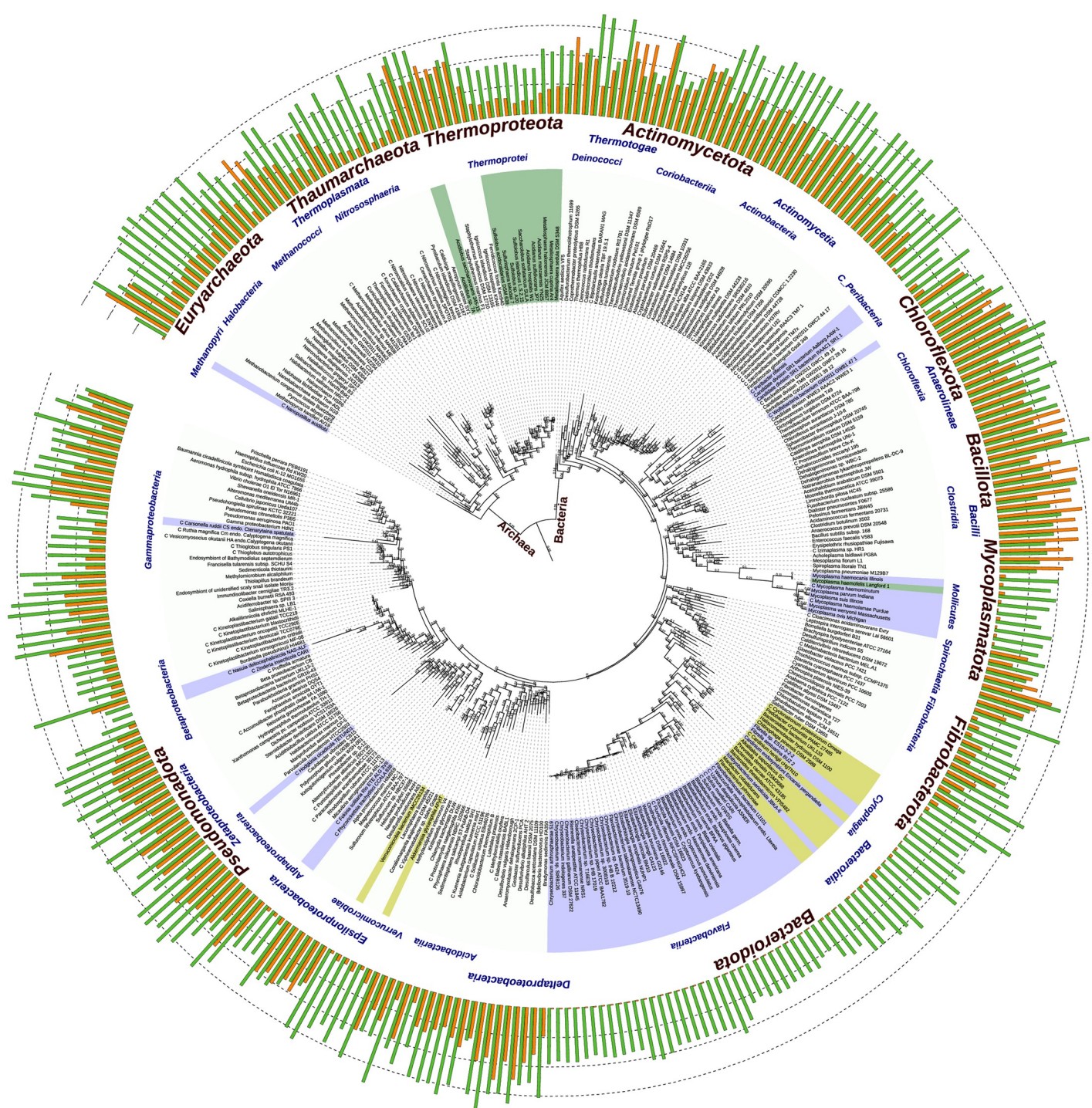

**Fig 9. Phylogenetic tree of representative organisms with different classes of antiSD/SD sequences.** To represent the phylogenetic distribution of organisms belonging to the different types of categorizations mentioned above, 323 representative members from our list of studied organisms were selected, and their 16S rRNA gene sequences were used to build the phylogenetic tree. The green and orange bars show the GC content and the percentage of mRNAs capable of establishing stable 16S rRNA/mRNA interactions, respectively. The names of the organisms are colored by one of the four groups to which they belong: colorless, yellow, green, and purple, to represent the groups i, ii, iii, and iv, respectively.

Table presents the complete list of 327 organisms in this classification, representing 5% of our study's organisms. Table 2 includes representative examples of this group's members.

**iii) "SD-Abundant" organisms lacking a conserved CCUCC.** This group of organisms is characterized by a modification in the CCUCC motif of their 16S rRNAs; nonetheless, their 16S rRNA 3'-termini can still form stable interactions with a significant fraction of the organism's total mRNAs, which is represented by orange bars in Fig 9 and might vary from 9.5% to more than 85% (Table 3). Members of this group are represented by the green color in Fig 9. They mainly belong to the domain Archaea, as are the cases of *Acidilobus* sp.7A and *Sulfolobus islandicus*, whose 16S rRNA genes present changes in their 3'-terminus, from the canonical CCUCC to CCUCU and CCUCA, respectively (Fig 10). The variant CCUCA appears in 36 out of 39 members of this category, all of which are predicted to use SD:antiSD for translation initiation. However, the adenine in this alternative core does not pair frequently with the SD, and is not part of the antiSD profile in 25 out of the 36 organisms. In exceptional cases, we also found bacterial organisms that belong to this group, like the Alphaproteobacteria *Erythrobacter litoralis*. In this case, the canonical CCUCC has changed to ACUCC (Fig 10). It would be expected that changes in the antiSD core would be accompanied by matching changes in the SD of many genes. That is the case for the C → U at the -4 position in the 16S rRNA of *Acidilobus* sp.7A that seems to have a corresponding G → A in the SD sequences of their mRNAs, as our thermodynamic analysis can tell. Interestingly, neither the C → A at the -4 position nor the C → A at -10 position in *Sulfolobus islandicus* and *Erythrobacter litoralis*, respectively, seems to have an interacting counterpart in the leader regions of their mRNAs. In addition to the organisms mentioned above, our study identified 36 organisms that belong to this classification (S2 Table). Table 3 includes representative examples of this group's members.

**iv) "SD-Scarce" organisms lacking a conserved CCUCC.** Organisms with these characteristics have been identified in the domain of Archaea, particularly in the Nanoarchaeia class, where many microorganisms are characterized by the absence of a leader region in their mRNAs [1]; consequently, the selective pressure to preserve a functional CCUCC in their 16S rRNA genes has been lost. In addition, we also identified organisms belonging to this classification in the bacteria domain, mainly in the Flavobacteriia class, which is consistent with previous reports [45, 46]. The 75 Flavobacteriia present in this category make up 67% of its total members, and they constitute 55% of all Flavobacteriia in our study. Our study identified 112 organisms in this classification (S2 Table), some of them are shown as purple-colored nodes in Fig 9. These results provide evidence that the loss of functional antiSD sequences is not restricted to archaea but might have happened at different points during the evolution of prokaryotic organisms. Table 4 includes representative examples of this group's members.

Special attention has been given to identifying prokaryotic organisms that do not possess the CCUCC at the 3' termini of their 16S rRNA genes [45, 46]. Our study has classified these organisms into our third and fourth categories based on thermodynamic analyses that use the base pair interactions between their 16S rRNA/mRNA to determine translation initiation. In our study, we conducted rigorous multiple sequence alignments of all the 16S rRNA 3'-termini previously identified in our motif sequence conservation analysis. As a result, we identified 96 bacterial and 39 archaeal new organisms whose 16S rRNA genes lack the canonical antiSD sequence. These new cases, along with the previously reported ones, are shown in the S4 Table. It is important to note that although genomic sequencing errors can lead to a false positive in our organism list, two factors make it unlikely for this to occur in the found sequences. First, the number of 16S rRNA gene copies: almost half of these organisms have more than one copy but maintain the same antiSD variant between them. Second, the preservation of inter-species changes of sequence: the fact that different genomes of closely related species present the same antiSD sequence also occurs in the vast majority of them. This set of organisms includes five

**Table 2. "SD-Scarce" organisms with conserved CCUCC.**

| D | Phylum | Class | Species | % SD | Total |
|---|---|---|---|---|---|
| Bacteria | Bacteroidota | Bacteroidia | *Alistipes finegoldii* | 2 | 66 |
| | | | *Bacteroides fragilis* | 2 | |
| | | | *Prevotella oris* | 2 | |
| | | | . . . | | |
| | | Chitinophagia | *Arachidicoccus B3-10* | 1 | 19 |
| | | | *Chitinophaga caeni* | 2 | |
| | | | *Niastella koreensis* | 1 | |
| | | | . . . | | |
| | | Cytophagia | *Bernardetia litoralis* | 0.1 | 54 |
| | | | *Hymenobacter nivis* | 2 | |
| | | | *Spirosoma aureum* | 1 | |
| | | | . . . | | |
| | | Flavobacteriia | *Aquimarina sp. AD1* | 1 | 130 |
| | | | *Flavobacterium album* | 1 | |
| | | | *Gramella fulva* | 1 | |
| | | | . . . | | |
| | | Saprospiria | *Saprospira grandis* | 2 | 2 |
| | | | *Haliscomenobacter hydrossis* | 1 | |
| | | Sphingobacteriia | *Pedobacter cryoconitis* | 1 | 31 |
| | | | *Sphingobacterium mizutaii* | 2 | |
| | | | . . . | | |
| | Cyanobacteria | Cyanophyceae | *Prochlorococcus marinus* | 3 | 4 |
| | Pseudomonadota | Gammaproteobacteria | *Endosymbiont of S. gigas* | 1 | 1 |
| | Mycoplasmatota | Mollicutes | *Mycoplasma genitalium* | 2 | 6 |
| | | | . . . | | |
| | Verrucomicrobiota | Verrucomicrobiae | *Akkermansia glycaniphila* | 4 | 2 |
| | | | . . . | | |
| **Archaea** | Nanoarchaeota | Nanoarchaeales | *Nanoarchaeum equitans* | 3 | 1 |

Identification of the CCUCC at the 3'-termini of the 16S rRNAs of the studied organisms was made using the MAST program and the MEME matrices as described in the Material and Methods sections. The number of mRNAs capable of establishing stable interactions with the 16S rRNAs is indicated as a percentage. Only representative organisms per phylogenetic class are included in the table, but the total number of organisms identified in each category is shown in the last column. In the table header, "D" stands for domain.

**Table 3. "SD-Abundant" organisms lacking a conserved CCUCC.**

| D | Phylum | Class | Species | % SD | Total |
|---|---|---|---|---|---|
| **Archaea** | Thermoproteota | Thermoprotei | *Acidianus sulfidivorans* | 14 | 38 |
| | | | *Acidilobus saccharovorans* | 54 | |
| | | | *Acidilobus sp. 7A* | 53 | |
| | | | *Metallosphaera prunae* | 26 | |
| | | | *Sulfolobus islandicus* | 18 | |
| | | | . . . | | |
| **Bacteria** | Pseudomonadota | Alphaproteobacteria | *Erythrobacter litoralis* | 43 | 1 |

In the table header, "D" stands for domain.

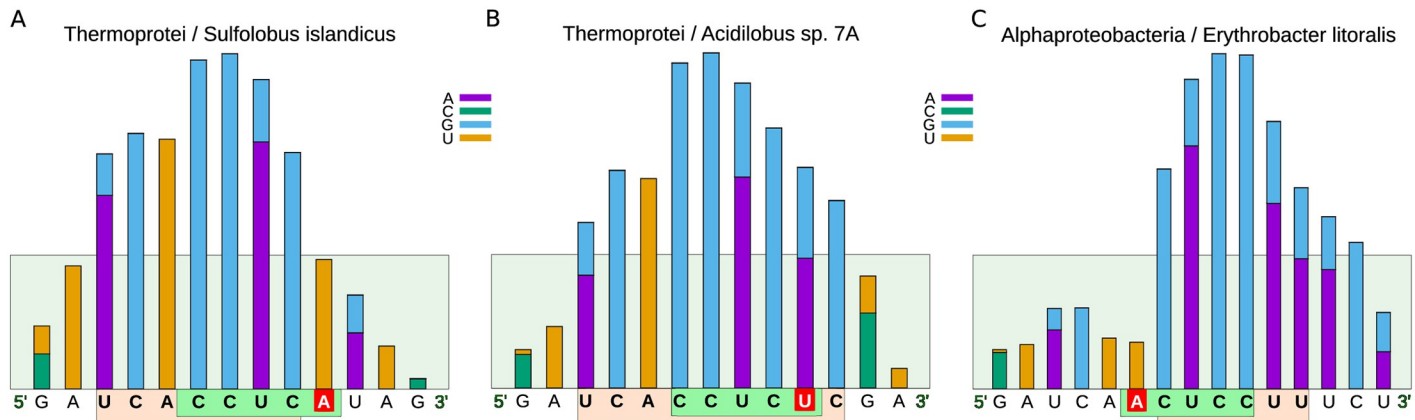

**Fig 10. Histograms of the frequencies at which the bases of the 3'-terminus of the 16S rRNA interact with the 5' upstream region of the mRNAs with relatively high frequencies for organisms lacking the canonical `CCUCC`.** A) and B) correspond to histograms of archaeal phylogenetic organisms, while C) corresponds to the Alphaproteobacteria class of the Bacteria domain. According to the frequencies of interaction with the leader regions of the mRNAs, the bases at the 3' end of the 16S rRNA were defined as part of the antiSD sequence and are indicated in brown boxes. The canonical sequences `CCUCC` of the 16S rRNA are framed in green boxes, and the changes that move them away from the consensus are indicated in red boxes.

different phyla (Bacteroidota, Pseudomonadota, Candidatus Absconditabacteria, Candidatus Wolfebacteria, and Candidatus Campbellbacteria) and five different classes (Bacteroidia, Cytophagia, Flavobacteriia, Alphaproteobacteria, and Betaproteobacteria) in the case of bacterial organisms and three different phyla (Thermoproteota, Nanoarchaeota, and Crenarchaeota) and two different classes (Thermoprotei, C. Nanoarchaeia) in Archaea. It has been reported that *Sulfolobus solfataricus* does not conserve the `CCUCC` core [16]. Our study confirms this observation in Archaea, identifying 39 species with this characteristic as well. Due to inaccurate genomic annotations or failures in computational algorithms, it is important to consider that the lack of in silico identification of antiSD elements in a genomic sequence does not necessarily imply the genuine absence of the antiSD sequence. For example, from the set of organisms lacking the `CCUCC` reported in [46], we identified three of them (Bacillus azotoformans LMG 9581 (GCA_000307855.1), Bacillus bataviensis LMG 21833 (GCA_000307875.1), and Microcystis aeruginosa SPC777 (GCA_000412595.1)) that do have a canonical antiSD sequence in their corresponding 16S rRNAs.

## Conclusions

Over the past 50 years, our understanding of the elements defining translation initiation in prokaryotic organisms has evolved significantly since J. Shine and L. Dalgarno's groundbreaking discovery of a consensus sequence at the 3' end of 16S rRNA and its corresponding sequence in the leader region of mRNAs, to the view that we have today thanks to the results obtained by high-throughput techniques and the analysis of thousands of fully-sequenced genomes. In general terms, we divided the prokaryotic organisms into two groups: those that use, to a significant extent, the 16S rRNA/mRNA base-pairing interactions to define the translation initiation of their mRNAs and those that are mostly independent of these interactions.

Regarding the first of these groups, our results expand previous reports on variations in the SD sequence observed between different phylogenetic groups, mainly observed when comparing SD consensus sequences from bacteria and archaea [4, 47]. Our analysis of nearly 6,500 organisms belonging to 87 phylogenetic classes allowed us to define fifteen additional antiSD consensuses that kept the `CCUCC` unchanged but exhibited particular changes around this

**Table 4. "SD-Scarce" organisms lacking a conserved CCUCC.**

| D | Phylum | Class | Species | % SD | Total |
|---|--------|-------|---------|------|-------|
| **Archaea** | Nanoarchaeota | C.Nanoarchaeia | *C. Nanopusillus acidilobi* | 7 | 1 |
| **Bacteria** | Bacteroidota | Flavobacteriia | *Weeksella virosa* | 2 | 75 |
| | | | *Chryseobacterium glaciei* | 3 | |
| | | | *Riemerella anatipestifer* | 3 | |
| | | | . . . | | |
| | | Bacteroidia | *Alistipes_sp._dk3624* | 7 | 1 |
| | | Cytophagia | *Fibrella sp.ES10-3-2-2* | 3 | 3 |
| | | | *Fibrella aestuarina* | 5 | |
| | | | *Cardinium sp.cEper1* | 5 | |
| | Mycoplasmatota | Mollicutes | *Mycoplasma haemocanis* | 8 | 10 |
| | | | *Mycoplasma ovis* | 5 | |
| | | | . . . | | |
| | Pseudomonadota | Alphaproteobacteria | *C. Hodgkinia cicadicola* | 3 | 6 |
| | | | *C. Fokinia solitaria* | 7 | |
| | | | . . . | | |
| | | Betaproteobacteria | *C. Vidania fulgoroideae* | 3 | 3 |
| | | | *C. Nasuia deltocephalinicola* | 2 | |
| | | | . . . | | |
| | | Gammaproteobacteria | *C. Carsonella ruddii CE* | 2 | 7 |
| | | | *C. Carsonella ruddii PC* | 3 | |
| | | | . . . | | |
| | Verrucomicrobiota | No rank | *C. Pinguicoccus_supinus* | 3 | 1 |
| | Candidatus Absconditabacteria | No rank | *C. division SR1 RAAC1* | 2 | 2 |
| | | | *C. division SR1 Aalbo* | 7 | |
| | Candidatus Campbellbacteria | No rank | *C. Campbellbacteria* | 5 | 1 |
| | Candidatus Peregrinibacteria | No rank | *C. Peribacter riflensis* | 7 | 1 |
| | Candidatus Wolfebacteria | No rank | *C. Wolfebacteria GW2* | 2 | 1 |

Representative examples of organisms belonging to class iv are shown. The complete list of organisms in this category is listed in S2 Table. "C." in species names stands for "Candidatus". In the table header, "D" stands for domain.

core element that were associated with the phylogenetic origin of the organisms. The existence of this set of consensus sequences is consistent with the idea of a gradual evolution of the antiSD rather than an abrupt binary change between the antiSD sequences of the two great phylogenetic domains, bacteria and archaea. This idea is further supported by our finding that Group 12 of our antiSD classification has both, bacterial and archaeal members. In addition to the fifteen different consensus sequences of antiSD, we identified clear examples of organisms in which 16S rRNAs have nucleotide changes in one of the CCUCC bases of this almost universal element of the 16S rRNAs in prokaryotic organisms. Interestingly, in some cases (for instance, *Acidilobus* sp. 7A), we identified a compensatory nucleotide change in most of the mRNA leader sequences in such a way that the degree of 16S rRNA/mRNA interactions remained similar to the one observed in most of the organisms that belong to any of our fifteen antiSD categories, while in other cases, the change in the CCUCC excluded the altered position from interacting.

Our thermodynamic analysis of the 16S rRNA/mRNA interactions also allowed us to identify organisms whose translation initiation is independent of the nucleotide interaction of these two RNA molecules. Our results are in good agreement with the previously reported

leaderless mRNA in many archaea and some other bacterial organisms of the Flavobacteriia class, where the lack of the CCUCC in the 16S rRNA 3'-termini regions has been reported. Additionally, we have substantially expanded this list by nearly doubling it, discovering representative organisms from five previously unexplored phyla and five distinct classes. This finding supports the notion that the loss of the translation initiation pathway, reliant on 16S rRNA/mRNA base-pairing interactions, might occur through multiple evolutionary events.

Collectively, our results illuminate the rich diversity of translation initiation strategies across the prokaryotic domain, as well as the incredible adaptability and diversity of life at the molecular level.

## Supporting information

**S1 Table. List of the 6,457 species included in the study.** The table includes information on the species identifier in the KEGG database and its taxonomy annotations.
(XLSX)

**S2 Table. Summary of binding results for all organisms.** The table includes the following information: sequence of the unstructured region in the 16S rRNA 3'-terminus, sequence of the antiSD, percentage of genes with a binding SD, and the binding **Category** of the organism. Note: antiSD sequence is not shown if fewer than 10% of the genes showed binding, stated as no-antiSD.
(XLSX)

**S3 Table. List of the 323 species included in the phylogeny.** The table includes information on the species identifiers in the KEGG database and their taxonomy annotations.
(XLSX)

**S4 Table. List of the 266 prokaryotic organisms lacking the antiSD core CCUCC as of today.** The table includes information on the species name, the last bases of the 16S rRNA, the assembly accession number, the taxonomy annotations until the class level, and a column "Reference" indicating if it was previously reported or if it is a species reported in our study.
(XLSX)

**S1 Fig. Conserved motifs in the 16S rRNA identified using the MEME software.** The MEME matrix shows all the non-variable regions of the 16S rRNA, which commonly includes the CCUCC in the last motif.
(PDF)

**S2 Fig. The average of the antiSD profile is compared to the average position of SDs in an organism.** Linear regression indicates that the correlation is poor.
(PNG)

## Acknowledgments

We sincerely thank Sonia Dávila, Armando Hernández, and Cinthia Nuñez for the continuous supervision of the project with enriching academic debates and critical advice; Mabel Rodríguez and Ilse Salinas for their tireless efforts in assisting with the writing and data completion; Felix Santana for his invaluable support in R programming and Inkscape graphics software; Arturo Ocádiz and Juan Manuel Hurtado for computer support, and Shirley Ainsworth for bibliographical assistance.

## Author Contributions

**Conceptualization:** Karel Estrada, Alejandro Garciarrubio, Enrique Merino.

**Data curation:** Karel Estrada, Alejandro Garciarrubio, Enrique Merino.

**Formal analysis:** Karel Estrada, Alejandro Garciarrubio, Enrique Merino.

**Investigation:** Karel Estrada, Alejandro Garciarrubio, Enrique Merino.

**Methodology:** Karel Estrada, Alejandro Garciarrubio, Enrique Merino.

**Project administration:** Enrique Merino.

**Resources:** Enrique Merino.

**Software:** Karel Estrada, Enrique Merino.

**Supervision:** Enrique Merino.

**Validation:** Karel Estrada, Alejandro Garciarrubio, Enrique Merino.

**Visualization:** Karel Estrada, Alejandro Garciarrubio, Enrique Merino.

**Writing – original draft:** Karel Estrada, Alejandro Garciarrubio, Enrique Merino.

**Writing – review & editing:** Karel Estrada, Alejandro Garciarrubio, Enrique Merino.

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
