## [Decision Letter · Decision Letter 0]

23 May 2023

PONE-D-23-13112Unraveling the Plasticity of Translation Initiation in Prokaryotic Organisms: Beyond the Invariant Shine-Dalgarno SequencePLOS ONE

Dear Dr. Merino,

Thank you for submitting your manuscript to PLOS ONE. After careful consideration, we feel that it has merit but does not fully meet PLOS ONE’s publication criteria as it currently stands. Therefore, we invite you to submit a revised version of the manuscript that addresses the points raised during the review process.

ACADEMIC EDITOR: After careful reading the manuscript and observing the reviewers' independent comments, my opinion is that this manuscript has some serious concerns, must be addressed through a major revision and reframe the findings as per the reviewers' suggestions.Re-run the analysis using a larger upstream window and make sure to explain the changes in the results/hypothesis the authors infer from this study.Reviewer 2 has several major concerns, that leads them to recommend a rejection of the manuscript. I would take those issues seriously and address all the points carefully throughout the manuscript.My concern is also with the structure of the introduction section and the discussion section. These sections must be rewritten in a lucid flow and the discussion should tally pointwise/straight-forward with the findings presented in the data. Please submit your revised manuscript within 40 days from now. If you will need more time than this to complete your revisions, please reply to this message or contact the journal office at plosone@plos.org. Please include the following items when submitting your revised manuscript:A rebuttal letter that responds to each point raised by the academic editor and reviewer(s). You should upload this letter as a separate file labeled 'Response to Reviewers'.A marked-up copy of your manuscript that highlights changes made to the original version. You should upload this as a separate file labeled 'Revised Manuscript with Track Changes'.An unmarked version of your revised paper without tracked changes. You should upload this as a separate file labeled 'Manuscript'.

We look forward to receiving your revised manuscript.

Kind regards,

Tarunendu Mapder, Ph.D.

Academic Editor

PLOS ONE

Journal Requirements:

Reviewers' comments:

Reviewer's Responses to Questions

**Comments to the Author**

1. Is the manuscript technically sound, and do the data support the conclusions?

Reviewer #1: Yes

Reviewer #2: No

Reviewer #3: Yes

2. Has the statistical analysis been performed appropriately and rigorously? 

Reviewer #1: Yes

Reviewer #2: No

Reviewer #3: N/A

3. Have the authors made all data underlying the findings in their manuscript fully available?

Reviewer #1: Yes

Reviewer #2: Yes

Reviewer #3: Yes

4. Is the manuscript presented in an intelligible fashion and written in standard English?

Reviewer #1: Yes

Reviewer #2: Yes

Reviewer #3: Yes

5. Review Comments to the Author

Reviewer #1: The manuscript by Estrada et al. reports a thorough computational survey of SD-anti SD interactions in prokaryotes based on the analysis of 6,457 genomes from both Bacteria and Archaea. As compared to previous studies, this work includes the computation of free energy estimates for the proposed SD-anti SD pairings. The authors have analyzed the sequences of the 3' extremities of the 16S rRNA genes, together with the sequences located upstream from the annotated start codons. Most anti SD sequences are built around the classical CCUCC core. In a first part, the authors make a classification of organisms (that use SD in more than 10% of their genes) regarding the overall anti SD sequence and the position of the CCUCC core within this sequence. In a second part, the authors classify all studied organisms into four classes, depending on the presence or absence of an anti SD sequence carrying the CCUCC core, and on the occurrence of possible base pairing of the 16S rRNA with the mRNA regions upstream from the start codon. The results are finally discussed in order to get insights into the evolution of translation initiation.

Overall, this is a very interesting and comprehensive study, adequately conducted and clearly presented. As it is the case for this kind of studies, possible sources of inaccuracies are linked to errors in genome annotations. Indeed, it is required that the 3' end of the 16S rRNA is correctly identified and that the start codon on the mRNA is correctly annotated. Here, the authors have used MEME motifs which, in my understanding, allow to evaluate the presence of the CCUCC sequence whatever the mapping of the 3' end. This is discussed at the end of the results section. Because this is a question that may worry the reader, it may be useful to briefly mention in the Materials and Methods section (Identification of the 16S rRNA 3'-termini) that particular attention was paid to cases where CCUCC was not found. Regarding annotation of start codons, it is unclear to me whether the authors solely relied on annotated start codons or whether they have considered possible neighboring in-frame alternative start codons. If alternative start codons have not been considered, which would be understandable, it would be worth mentioning that this is a possible source of inaccuracy.

Following are some more specific suggestions that the authors may wish to consider:

- Lines 305-312: Are the variations in the position of CCUCC related to variable positions of the SD sequence on the mRNA with respect to the start codon?

- Is there a correlation between the mean delta G of SD-anti SD pairing and the optimal growth temperature of the considered organism?

- Line 418: the statement that such organisms initiate translation independently of antiSD/SD interactions may appear a little bit confusing. Because the results are statistical, it cannot be excluded that the translation of some mRNAs relies on pairing with the 3'-end of 16S rRNA. Thus, "mostly independently" may be more appropriate.

- Ref 35 seems to be incomplete.

Reviewer #2: Merino and coworkers looked at potential SD-ASD interactions for a large number (>6000) of bacteria and archaea. They first used a MEME/MAST approach to identify and extract the 3’ portion 16S rRNA sequence. Then, they used RNAfold to assign the single-stranded 3’ tail (13-15 nt, depending on the organism). Then, they grabbed the 8 nt sequence (positions -2 to -10) upstream of each start codon. Per genome, these 8 nt TIR sequences were annealed in silico to the 3’ tail of 16S rRNA using RNAhybrid, and interactions more stable than -8.4 kcal/mol were deemed SD-ASD interactions. Histograms were then made, which showed the frequency of pairing for each 16S rRNA position across the tail sequence. Histograms representing various clades were compared to assess differences.

While the general question of how mRNA-rRNA pairing is used (and varies) among numerous prokaryotes is an interesting one, there are major problems with this study.

1. The sequence from -2 to -10 will contain only a portion of the SD in many cases. The optimal spacing between the SD and start codon is 6-8 nt (PMID 1375309), so the 5’ portion of the SD extends well upstream of -10 for many genes. Thus, the authors need to re-run their analysis using a larger upstream window (-2 to -20, minimally) in order to capture all SD-ASD interactions and report accurate pairing frequencies at each position.

2. How the authors assigned length of 3’ tail of 16S rRNA is unclear, and the use of for example 14 nt in some cases and 16 nt in others introduces an arbitrary variable which weakens the comparative approach. The 3’ tail has been experimentally determined for many bacteria (PMID 34812116), which the authors fail to acknowledge. E. coli 16S rRNA ends with A1542 (contrary to Fig. 1), while 16S of many other bacteria ends with nt 1544. It would be reasonable to use 1530-1544 for all organisms, but if they choose to do so they should also clearly explain the rationale and caveats.

3. The authors seem unaware of major advances in our understanding of Bacteroidia ribosomes in recent years (PMID 33330920, PMID 34966783, PMID 36727479). Fredrick and coworkers have shown that the 3’ tail of 16S rRNA is sequestered on the 30S platform, in a pocket formed by S21, S18, and S6. This explains why ribosomes of these organisms are “blind” to SD sequences. It was also shown that certain ribosomal protein genes in these organisms actually do have SD sequences, which act in regulation of translation. Examples of alternative ASD sequences in certain Flavobacteria were pointed out, and in all cases, the fully complementary SD sequence is seen upstream of rpsU. This natural covariation underscores the importance of SD-ASD pairing in translation of at least one gene—rpsU. This published work needs to be appropriately discussed and cited by Merino and coworkers.

4. In Table S2, the authors assign “no antiSD” to organisms which do not use SD sequences (e.g., Bacteroidia). But this is incorrect, because these organisms do have an ASD which is normally sequestered on the platform domain. The authors need to use more precise language throughout the text, figures, and tables.

5. The authors use a threshold of -8.4 kcal/mol to call a SD-ASD interaction. Why was this value chosen? No details are provided, just a citation.

6. Fig. 9 shows relative G:C content across representative organisms, but G:C content is NOT taken into account when %SD is calculated. Consequently, the %SD values reported are overestimates. When G:C content is taken into account, SD occurrences are lower than expected by chance for the Bacteroidia (PMID 20308567, PMID 33330920, PMID 36727479).

7. The authors claim to be the first to report an alternative SD-ASD interaction the archaea. But, Tolstrup et al. 2000 (PMID 10879562) showed that Sulfolobus sofataricus uses unique SD sequences due to a substituted ASD, work that has been well cited since.

Minor issues:

1. The term “archaebacteria” is outdated; use “archaea” instead.

2. Figure 2 seems to be missing some components.

Reviewer #3: Efficient and correct translation initiation in many prokaryotes rely on specific base pairing between mRNA (the SD sequences) and ribosomal rRNA (the antiSD sequences). However, such SD:antiSD interactions are not universally conserved, in terms of sequence identity, length, and position. In this manuscript, the authors surveyed more than 6400 genomes from bacteria and archaea and analyzed potential SD:antiSD interactions for each of them. The results provide a systematic view on these interactions and their variability and flexibility. These are valuable information and will be beneficial to those who are interested in this field. Below are some related concerns.

The Introduction section is important for readers to gain an idea about what has been known in the field and how the current research is related to it. In this regard, I think the current Introduction still has some room to improve. In addition, the first paragraph (lines 194-214) in Results and Discussion reads like an introduction, and thus I suggest to move it to Introduction.

Minor points:

The term “UTR” is not defined.

Line 32: “the region preceding the ATG…”, “AUG” should be used instead.

Lines 197-198: “a sequence in the mRNA located 5 to 8 nucleotides from the start codon…”, the direction (upstream) should be included here.

Lines 216-219: As leaderless mRNAs are not common in eukaryotes, I am not sure why the authors stated that eIF2 is crucial for start codon selection. Also, the cited references are not related to eIF2.

Line 237: Does the “rightmost” mean most downstream?

Lines 276-277: Does “the interacting bases” indicate canonical and wobble base pairs?

In Fig. 1, IF1 and IF3 are not positioned correctly. The authors may consider to remove them from the figure since they are not relevant.

In Fig. 4, 5, 7, 8, and 10, the legend “T” should be replaced by “U”.

The DOI in reference #23 does not seem to be a valid link.

6. PLOS authors have the option to publish the peer review history of their article (what does this mean?). If published, this will include your full peer review and any attached files.

Reviewer #1: No

Reviewer #2: No

Reviewer #3: No

---

## [Author Response · Author response to Decision Letter 0]

3 Jul 2023

PONE-D-23-13112

Response to the Editor:

> Dear Editor, all your indications have been addressed, including updating the results for the larger UTR window, paying careful attention to all reviewer’s comments (especially Reviewer #2), and strengthening the Introduction and Discussion.

> The new UTR size modified our results in some cases, nevertheless, in all cases the difference in their values were minimal. Our main conclusions remain. We believe that the new analysis is more robust.

> Due to extensive rewriting, the "Revised_Manuscript_with_Track_Changes.docx" is very difficult to read. We want to mention the most important modifications:

> 1) We updated all information that changed with the new UTR from -20 to -2 before the AUG.

> 2) The number of different antiSD profiles changes from 10 to 15.

> 3) That required changing some values in all tables, especially Table 1.

> 4) By your suggestion and that of the reviewers, we wrote a new Introduction.

> 5) A large section of Results and Discussion was removed and integrated in the Introduction.

Reviewer #1: 

The manuscript by Estrada et al. reports a thorough computational survey of SD-anti SD interactions in prokaryotes based on the analysis of 6,457 genomes from both Bacteria and Archaea. As compared to previous studies, this work includes the computation of free energy estimates for the proposed SD-anti SD pairings. The authors have analyzed the sequences of the 3' extremities of the 16S rRNA genes, together with the sequences located upstream from the annotated start codons. Most anti SD sequences are built around the classical CCUCC core. In a first part, the authors make a classification of organisms (that use SD in more than 10% of their genes) regarding the overall anti SD sequence and the position of the CCUCC core within this sequence. In a second part, the authors classify all studied organisms into four classes, depending on the presence or absence of an anti SD sequence carrying the CCUCC core, and on the occurrence of possible base pairing of the 16S rRNA with the mRNA regions upstream from the start codon. The results are finally discussed in order to get insights into the evolution of translation initiation.

Overall, this is a very interesting and comprehensive study, adequately conducted and clearly presented. As it is the case for this kind of studies, possible sources of inaccuracies are linked to errors in genome annotations. Indeed, it is required that the 3' end of the 16S rRNA is correctly identified and that the start codon on the mRNA is correctly annotated. Here, the authors have used MEME motifs which, in my understanding, allow to evaluate the presence of the CCUCC sequence whatever the mapping of the 3' end. This is discussed at the end of the results section. Because this is a question that may worry the reader, it may be useful to briefly mention in the Materials and Methods section (Identification of the 16S rRNA 3'-termini) that particular attention was paid to cases where CCUCC was not found. Regarding annotation of start codons, it is unclear to me whether the authors solely relied on annotated start codons or whether they have considered possible neighboring in-frame alternative start codons. If alternative start codons have not been considered, which would be understandable, it would be worth mentioning that this is a possible source of inaccuracy.

The use of MEME has been further clarified in the Material and Methods section of the new version of our article. We also stress the attention paid to ensuring the presence/absence of the CCUCC motif. 

We believe that, in most cases, the boundaries of the genes defined by NCBI are correct and that the low frequencies of the incorrect annotations do not statistically affect the overall results and definitions of the Shine-Dalgarno sequences obtained in our analysis; therefore, we did not correct any possible annotation inaccuracies. This is now commented on in the Discussion section.

Following are some more specific suggestions that the authors may wish to consider:

- Lines 305-312: Are the variations in the position of CCUCC related to variable positions of the SD sequence on the mRNA with respect to the start codon?

> While the CCUCC is always at the same position, the antiSD profiles are shifted relative to it. So we investigated whether "variations in the position of the antiSD profile related to variable positions of the SD sequence on the mRNA with respect to the start codon?" Preliminary results indicate that the mean position of the antiSD profile in an organism is poorly correlated with the mean position of SD (r=0.06, see Fig Extra zero). 

> === Please find Fig Extra Zero in the PDF version. It was not possible to include in this text-format response =======

> Fig Extra Zero. The average of the antiSD profile is compared to the average position of SDs in an organism. Linear regression indicates that the correlation is poor.

- Is there a correlation between the mean delta G of SD-anti SD pairing and the optimal growth temperature of the considered organism?

> We thank reviewer #1 for this interesting question. The TEMURA database (http://togodb.org/db/tempura) records many prokaryotes' optimal growth temperature (OGT). It intersects with 1,239 of our species. Within these, there is a significant correlation of mean dG with OGT (Fig Extra 1). Interestingly, this is not mediated by genomic GC. Further analysis using multiple linear regression showed that dG is significantly correlated with OGT (P <0.001), when controlling for GC, while dG is not significantly correlated with GC (p=0.571), when controlling for OGT.

> We will be happily to include this result in the article if the Editor deems it convenient.

> === Please find Fig Extra 1 in the PDF version. It was not possible to include in this text-format response =======

> Fig. extra 1. The TEMPURA database (http://togodb.org/db/tempura) intersects with our genomes, encompassing 1239 species. Our analysis reveals a significant negative correlation between the mean dG of these species and their optimal growth temperature. This correlation proves to be more robust compared to the correlation observed with genomic GC content (refer to Fig. Extra 2). By performing a multiple linear regression analysis, we found that temperature was significantly associated with dG (β = -0.0444, p < 0.001), indicating that for each unit increase in temperature, dG decreased by 0.0444 units when controlling for GC content. However, GC content was not significantly associated with dG (β = 0.0021, p = 0.571) when controlling for temperature, indicating that the effect of temperature on dG is independent of GC content.

- Line 418: the statement that such organisms initiate translation independently of antiSD/SD interactions may appear a little bit confusing. Because the results are statistical, it cannot be excluded that the translation of some mRNAs relies on pairing with the 3'-end of 16S rRNA. Thus, "mostly independently" may be more appropriate.

> We now qualify our statements with ‘mostly’, wherever necessary.

- Ref 35 seems to be incomplete.

> The title, "How changes in anti-SD sequences would affect SD sequences in Escherichia coli and Bacillus subtilis", was incomplete. This has been corrected.

Reviewer #2:

Merino and coworkers looked at potential SD-ASD interactions for a large number (>6000) of bacteria and archaea. They first used a MEME/MAST approach to identify and extract the 3’ portion 16S rRNA sequence. Then, they used RNAfold to assign the single-stranded 3’ tail (13-15 nt, depending on the organism). Then, they grabbed the 8 nt sequence (positions -2 to -10) upstream of each start codon. Per genome, these 8 nt TIR sequences were annealed in silico to the 3’ tail of 16S rRNA using RNAhybrid, and interactions more stable than -8.4 kcal/mol were deemed SD-ASD interactions. Histograms were then made, which showed the frequency of pairing for each 16S rRNA position across the tail sequence. Histograms representing various clades were compared to assess differences.

While the general question of how mRNA-rRNA pairing is used (and varies) among numerous prokaryotes is an interesting one, there are major problems with this study.

1. The sequence from -2 to -10 will contain only a portion of the SD in many cases. The optimal spacing between the SD and start codon is 6-8 nt (PMID 1375309), so the 5’ portion of the SD extends well upstream of -10 for many genes. Thus, the authors need to re-run their analysis using a larger upstream window (-2 to -20, minimally) in order to capture all SD-ASD interactions and report accurate pairing frequencies at each position.

> We made the unfortunate mistake of stating in Materials and Methods that “we considered the region of 8 nucleotides long located at the -10 and -2 bases upstream of the start codon”. In fact, we used 10 bases, from –14 to –4. 

> By the indication of the Editor and the very pertinent comment of Reviewer #2, we have re-run the whole analysis, searching for SD from –20 to –2, using a sliding window of 10 nt. In the new version of our article, the best match (most negative dG) from the 9 windows for each gene was used. While some details changed, the main conclusions of the work remained true.

> Corrections have been made all over the manuscript to reflect the new data, and the strategy is described in Materials and Methods.

2. How the authors assigned length of 3’ tail of 16S rRNA is unclear, and the use of for example 14 nt in some cases and 16 nt in others introduces an arbitrary variable which weakens the comparative approach. The 3’ tail has been experimentally determined for many bacteria (PMID 34812116), which the authors fail to acknowledge. E. coli 16S rRNA ends with A1542 (contrary to Fig. 1), while 16S of many other bacteria ends with nt 1544. It would be reasonable to use 1530-1544 for all organisms, but if they choose to do so, they should also clearly explain the rationale and caveats.

> We are thankful to Reviewer #2 for the careful review and criticism.

> We agree that the E. coli 16S rRNA 3` of Fig 1 and Fig 4 of the original version of our article was wrong. We have corrected these figures in the new version of our article.

> We created "Ribbon" (https://github.com/kjestradag/ribbon_1.2), to identify the 3' end of the 16S rRNAs, as part of a project to curate 16S RNAs, and it has proved its worth by discovering hundreds of annotation mistakes in NCBI annotations (unpublished data). The unstructured regions (UR) in the current work are extremely consistent: 99.9% of them have the same 5'-end nucleotide as verified with a Mafft alignment. Of those with the conserved CCTCC, 99.9% have it in the exact same position. All Bacteria also share the same 3' end, which gives a 15 nt long UR. On the other hand, all Archaea have a 2 nt shorter 3'-end, which gives a 13 nt long UR. The difference between Bacteria and Archaea is biologically relevant: the signature for the conserved region is 2 nt shorter in Archaea. It should be noted that the most common 3'-end in NCBI annotated 16S sequences is 2 nt shorter than ours, both for Bacteria and Archaea. It should also be noted that a significant number of them are erroneously missing the conserved 'CCTCC'. 

3. The authors seem unaware of major advances in our understanding of Bacteroidia ribosomes in recent years (PMID 33330920, PMID 34966783, PMID 36727479). Fredrick and coworkers have shown that the 3’ tail of 16S rRNA is sequestered on the 30S platform, in a pocket formed by S21, S18, and S6. This explains why ribosomes of these organisms are “blind” to SD sequences. It was also shown that certain ribosomal protein genes in these organisms actually do have SD sequences, which act in regulation of translation. Examples of alternative ASD sequences in certain Flavobacteria were pointed out, and in all cases, the fully complementary SD sequence is seen upstream of rpsU. This natural covariation underscores the importance of SD-ASD pairing in translation of at least one gene—rpsU. This published work needs to be appropriately discussed and cited by Merino and coworkers.

> We are sorry for the omission. The research on Flavobacterium johnsoniae and other Bacteroidia is extremely interesting and relevant to our work. It also supports our finding of the near complete absence of the SD mechanisms in the whole Bacteroidota phylum. We have included this research in the Introduction and Discussion sections of our paper and included the relevant references. Flavobacterium johnsoniae is now included in the phylogeny shown in Fig 9 and commented on in the Discussion section of our article.

4. In Table S2, the authors assign “no antiSD” to organisms which do not use SD sequences (e.g., Bacteroidia). But this is incorrect, because these organisms do have an ASD which is normally sequestered on the platform domain. The authors need to use more precise language throughout the text, figures, and tables.

> In our study, we defined that an antiSD calculation would not be performed when the percentage of genes demonstrating a significant interaction was less than 10%. This criterion was consistently applied, even in cases where experimental evidence indicated the involvement of the SD mechanism in only a few genes, such as in the case of Bacteroidia. We made this decision and many others based on a universal criterion to ensure consistency throughout our analysis.

> While revising our analysis, we evaluated the impact of various percentage thresholds on the classification of organisms utilizing the SD:ASD mechanism. We discovered that, exclusively within the 4-7% range, any chosen threshold resulted in almost identical classifications, indicating a degree of stability. This suggests that any threshold within this range would be less arbitrary. Consequently, we believe a 5.5% threshold is more suitable than a 10% threshold.

> We agree with Reviewer #2 that “no antiSD” is a misnomer. We have changed the label to “mainly undetected” antiSD.

5. The authors use a threshold of -8.4 kcal/mol to call a SD-ASD interaction. Why was this value chosen? No details are provided, just a citation.

> Following Starmer, J., et al (2006), we used the average dG of the 4 tetramers that can match the conserved CCUCCUU 16S core: AAGG (-7.1 kcal/mol), AGGA (-8.0 kcal/mol), GGAG (-9.3 kcal/mol) and GAGG (- 9.1 kcal/mol), as reported by RNAhybrid. The average is –8.375, which, rounded, gave us –8.4 kcal/mol as our threshold. This is now explained in Materials and Methods.

> An analysis using “fake” UTRs showed that this threshold agrees perfectly with the dG value at which the frequency of "True" SD interactions equals the frequency of random interactions (See Fig Extra 2). Values to the right would include more false positives than true positives. Conversely, values to the left would exclude more true positives than false positives.

> === Please find Fig Extra 2 in the PDF version. It was not possible to include in this text-format response =======

> Fig. Extra 2. The density of dG values from Real UTRs and "Fake" UTRs. The left Gumbel function is used to model the distribution of the minimum values. It is a very accurate representation of the dG distribution. Both collections contained 10K samples obtained from all genomes. To enrich for true SDs, the Real collection was restricted to matches with at least 3 nucleotides in the antiSD core (CCTCC), and a distance from the core to the start codon between 8 to 11 nucleotides. Both curves intersect at dG == -8.4, which is the same as our threshold to distinguish SD from non-SD.

> The lack of experimental data to calibrate the threshold is a common issue affecting all research that uses predicted dG to identify SD:antiSD interactions. In one report (Ma, J., Campbell, A., and Karlin, S., 2002), the authors extensively discuss their chosen threshold (mainly based on the expected antiSD interactions). In Fig Extra 3, we evaluated which threshold would most accurately replicate their data (considering they used a different predictor and window) and compared it to ours. The best-fitting threshold was –8.9 kcal/mol. The minor discrepancy we found (4% more genes with SD when using our threshold of -8.4 kcal/mol) does not warrant changing the way we chose our threshold. 

> === Please find Fig Extra 3 in the PDF version. It was not possible to include in this text-format response =======

> Fig. Extra 3. We aimed to align a dG threshold with the SD frequencies reported by Karlin 2002 (Ma, J., Campbell, A. and Karlin, S., 2002. Correlations between Shine-Dalgarno sequences and gene features such as predicted expression levels and operon structures. Journal of Bacteriology, 184(20), pp.5733-5745). That report includes the most comprehensive list of SD frequencies from Eubacteria and Archaea that we could find. It also provides a detailed justification for their chosen threshold. We found that a dG threshold of less than -8.9 provides the best fit to their data, whereas our selected threshold (dG less than -8.4) predicts approximately 4% more genes with SD.

6. Fig. 9 shows relative G:C content across representative organisms, but G:C content is NOT taken into account when %SD is calculated. Consequently, the %SD values reported are overestimates. When G:C content is taken into account, SD occurrences are lower than expected by chance for the Bacteroidia (PMID 20308567, PMID 33330920, PMID 36727479).

> We concur that GC is an important variable. However, it is not easily taken into account. Our study observed a very weak correlation between GC content and dG across all genomes analyzed, which suggests that adjusting the SD threshold based on GC content might have a negligible impact (slope = -0.004, R = -0.024; see Fig Extra 4). 

> === Please find Fig Extra 4 in the PDF version. It was not possible to include in this text-format response =======

> Fig. Extra 4. When all species are considered, there is almost no correlation between mean dG and genomic GC. Thus, correcting for GC with this "universal" slope would be inadequate.

> When examining individual phylum, we sometimes observed a stronger correlation. For instance, in the case of Pseudomonadota, our largest represented phylum, there was a moderately negative correlation (slope = -0.039, R = -0.516; see Fig Extra 5). However, this pattern was not consistent across all phyla. Many phyla in our dataset were represented by a limited number of genomes and exhibited a narrow range of GC content. Consequently, any corrections based on GC content would only apply to a handful of the largest phyla, and such a selective adjustment could potentially introduce bias into our analysis. Thus, we decided to refrain from making these adjustments to maintain the generalizability of our results.

> === Please find Fig Extra 5 in the PDF version. It was not possible to include in this text-format response =======

> Fig. Extra 5. Within some large taxonomic groups, especially if they span an ample GC range, there is an evident correlation between mean dG and GC, as shown for the Pseudomonadota phyla. Unfortunately, applying a correction with a "per taxa'' slope would be impossible for many of the taxa in our study.

> The phenomena in Bacteroidia appears in our data as a very low prediction of SD. An additional analysis demonstrated that the distribution of dG values within the phylum Bacteroidota is indeed less negative (indicating weaker interactions) than would be expected by chance (see Fig Extra 6) even without any correction for GC. This trend seems to be a unique characteristic of the UTRs within Bacteroidota genomes. These findings underscore the complexity and diversity of genomic interactions and further emphasize the need for phylum-specific studies, such as those cited by the reviewer, to fully understand these phenomena. 

> We decided to keep the GC values in Fig 9 because readers might find it informative.

> === Please find Fig Extra 6 in the PDF version. It was not possible to include in this text-format response =======

> Fig. Extra 6. In the Bacteroidota phyla, the modes of dG values of the UTRs tend to be less negative (indicating weaker interactions) compared to what would be expected by chance, as shown by the modes of dG values obtained from "fake" UTRs, which are random sequences possessing the same GC content. The observed deviation hints at the possibility of selection acting upon the UTRs to prevent interaction with the 16S rRNA 3'-end.

7. The authors claim to be the first to report an alternative SD-ASD interaction the archaea. But, Tolstrup et al. 2000 (PMID 10879562) showed that Sulfolobus sofataricus uses unique SD sequences due to a substituted ASD, work that has been well cited since.

> We thank Referee #2 for directing our attention to this observation. We now cite this precedent in the discussion.

Minor issues:

1. The term “archaebacteria” is outdated; use “archaea” instead.

> This has been corrected.

2. Figure 2 seems to be missing some components.

> The figure is an exact recreation (as svg) of a real output of RNAhybrid. RNA hybrid reports the target sequence (the 16S end in our case) but not the whole query. That is confusing. We have updated the figure to a better example.

Reviewer #3:

Efficient and correct translation initiation in many prokaryotes rely on specific base pairing between mRNA (the SD sequences) and ribosomal rRNA (the antiSD sequences). However, such SD:antiSD interactions are not universally conserved, in terms of sequence identity, length, and position. In this manuscript, the authors surveyed more than 6400 genomes from bacteria and archaea and analyzed potential SD:antiSD interactions for each of them. The results provide a systematic view on these interactions and their variability and flexibility. These are valuable information and will be beneficial to those who are interested in this field. Below are some related concerns.

The Introduction section is important for readers to gain an idea about what has been known in the field and how the current research is related to it. In this regard, I think the current Introduction still has some room to improve. In addition, the first paragraph (lines 194-214) in Results and Discussion reads like an introduction, and thus I suggest to move it to Introduction.

> We have rewritten the Introduction extensively and moved those paragraphs from the discussion into the new Introduction. We are sincerely grateful for the time and effort you dedicated to reviewing our article and for helping us improve it with your insightful comments.

Minor points:

+ The term “UTR” is not defined.

> Corrected

+ Line 32: “the region preceding the ATG…”, “AUG” should be used instead.

> Corrected

+ Lines 197-198: “a sequence in the mRNA located 5 to 8 nucleotides from the start codon…”, the direction (upstream) should be included here.

> Corrected

+ Lines 216-219: As leaderless mRNAs are not common in eukaryotes, I am not sure why the authors stated that eIF2 is crucial for start codon selection. Also, the cited references are not related to eIF2.

> Although rare, mRNAs with very short 5’ regions before the start codon exist in many eukaryotes, including humans. And it has been proposed that the ability to translate leaderless mRNAs is a conserved feature throughout all domains of life. 

> We mentioned elF2 because it is essential to determining the start codon in eukaryotes (which lack an SD), and its Archeal homolog, aIF2, does something similar for leaderless mRNA. We have edited the text to be clearer.

> We have also corrected the references.

+ Line 237: Does the “rightmost” mean most downstream?

> Downstream is clearer. We have corrected this.

+ Lines 276-277: Does “the interacting bases” indicate canonical and wobble base pairs?

> Yes, it does. But only G:U, which is a property of RNAhybrid. We have added clarification.

+ In Fig. 1, IF1 and IF3 are not positioned correctly. The authors may consider to remove them from the figure since they are not relevant.

> Fig 1. has been updated, including this suggestion.

+ In Fig. 4, 5, 7, 8, and 10, the legend “T” should be replaced by “U”.

> Corrected

+ The DOI in reference #23 does not seem to be a valid link.

> We have corrected the DOI (https://zenodo.org/record/7796827)

---

## [Editor Report · Decision Letter 1]

31 Jul 2023

Unraveling the Plasticity of Translation Initiation in Prokaryotes: Beyond the Invariant Shine-Dalgarno Sequence

PONE-D-23-13112R1

Dear Dr. Merino,

We’re pleased to inform you that your manuscript has been judged scientifically suitable for publication and will be formally accepted for publication once it meets the following changes and all outstanding technical requirements.

Kind regards,

Tarunendu Mapder, Ph.D.

Academic Editor

PLOS ONE

Additional Editor Comments (optional):

1. Please include the average of the antiSD profile is compared to the average position of SDs in an organism data in the supplementary materials.

2. All the extra figures explaining the reviewers' concern could be included in the SI.

---

## [Editor Report · Acceptance letter]

13 Aug 2023

PONE-D-23-13112R1 

Unraveling the Plasticity of Translation Initiation in Prokaryotes: Beyond the Invariant Shine-Dalgarno Sequence 

Dear Dr. Merino:

I'm pleased to inform you that your manuscript has been deemed suitable for publication in PLOS ONE. Congratulations! Your manuscript is now with our production department. 

Kind regards, 

on behalf of

Dr. Tarunendu Mapder 

Academic Editor

PLOS ONE